# Explicit Representation of Mechanical Functions for Maintenance Decision Support

Mengchu Song [1,]*, Ilmar F. Santos [2], Xinxin Zhang [1], Jing Wu [1] and Morten Lind [1]

1    Department of Electrical and Photonics Engineering, Technical University of Denmark,
2800 Kongens Lyngby, Denmark; xinz@dtu.dk (X.Z.); jinwu@dtu.dk (J.W.); mli@dtu.dk (M.L.)
2    Department of Civil and Mechanical Engineering, Technical University of Denmark,
2800 Kongens Lyngby, Denmark; ilsa@dtu.dk
*    Correspondence: menso@dtu.dk

**Abstract:** Artificial intelligence (AI) has been increasingly applied to condition-based maintenance (CBM), a knowledge-based method taking advantage of human expertise and other system knowledge that can serve as an alternative in cases in which machine learning is inapplicable due to a lack of training data. Functional information is seen as the most fundamental and important knowledge in maintenance decision making. This paper first proposes a mechanical functional modeling approach based on a functional modeling and reasoning methodology called multilevel flow modeling (MFM). The approach actually bridges the modeling gap between the mechanical level and the process level, which potentially extends the existing capability of MFM in rule-based diagnostics and prognostics from operation support to maintenance support. Based on this extension, a framework of optimized CBM is proposed, which can be used to diagnose potential mechanical failures from condition monitoring data and predict their future impacts in a qualitative way. More importantly, the framework uses MFM-based reliability-centered maintenance (RCM) to determine the importance of a detected potential failure, which can ensure the cost-effectiveness of CBM by adapting the maintenance requirements to specific operational contexts. This ability cannot be offered by existing CBM methods. An application to a mechanical test apparatus and hypothetical coupling with a process plant are used to demonstrate the proposed framework.

**Keywords:** condition-based maintenance; functional modeling; mechanical functions; functional reasoning; diagnostics; prognostics; reliability-centered maintenance

## 1. Introduction

As modern industry becomes increasingly complex, a superior maintenance strategy is the key to preserve or restore safety, reliability, and availability [1]. Maintenance has evolved in order to respond to changing expectations. Historically, maintenance has been essentially corrective. Since downtime was not significant on the industrial scale in the past, maintenance was only required when failure of equipment had already occurred. The concept of preventive maintenance became relevant as system became highly mechanized and downtime became a serious issue. Traditional preventive maintenance is time-based, that is, maintenance is performed at fixed intervals, regardless of the health of the equipment. However, Nowlan and Heap [2] revealed that most system failures are random rather than age-related, which implies that some costs associated with time-based preventive maintenance may be unnecessary. This leads to the most optimized preventive maintenance strategy, i.e., condition-based maintenance (CBM), which mainly relies on the data of condition monitoring and suggests maintenance actions only when there is evidence showing a necessity.

CBM is enabled by prognostics and health management (PHM) [3], which is technically related to diagnostics and prognostics. Diagnostics help to detect and identify faults in

the monitored system, while prognostics refer to prediction of how long it will take until the detected fault will develop into functional failure [4]. Diagnostic and prognostic methods for CBM are, in general, developed separately. Depending on the type of available information, diagnosis methods can be divided into physical-model-based [5], expert-knowledge-based [6], and historical-data-driven [7] methods. Prognostics mainly focus on the so-called remaining useful life (RUL), which can be predicted by approaches with similar categories as diagnostics [1].

In recent years, artificial intelligence (AI) has been increasingly applied in CBM, which has been integrated into traditional physics models and statistical methods, showing improved performance in terms of both diagnostics and prognostics [1,4,8]. One type of AI method, machine learning techniques, especially those using artificial neural networks (ANNs), has increased in popularity [9]. However, it also encounters several challenges in practice, as outlined below.

- Machine learning is essentially data-driven. However, it is usually associated with a lack of available data for model training, making the training process difficult [4].
- Machine learning algorithms usually do not have physical explanations regarding the trained models [10].
- Most machine learning methods reported in the literature focus on the maintained items rather than adopting a system-wide perspective [11], which means that they insufficiently reflect maintenance requirements in specific operational contexts.

Knowledge-based AI harnessing human expertise and existing system knowledge can be considered an alternative [11]. It is argued that functional knowledge can play an important role in maintenance decision support. Function is the most fundamental concept in maintenance. Industrial standards define maintenance as "a set of actions intended to retain an item in, or restore it to, a state in which it can perform the required function" [12]. Function is also used as the core criterion to determine maintenance requirements, like which assets need preventive maintenance and which do not. This is the principle of the maintenance optimization method known as reliability-centered maintenance (RCM) [13]. Moreover, understanding how functions are decomposed inside each asset is the key to understanding how a functional failure is developed and where a monitored anomaly can lead, which are useful for CBM. Therefore, it is instructive to develop a systematic approach that can explicitly represent the functional structure of assets and use AI algorithms to generate insights with respect to maintenance.

In this paper, a formal functional modeling methodology called multilevel flow modeling (MFM) [14] is applied to maintenance decision support. MFM provides a comprehensive framework to map between functions, behaviors, and structures. Most importantly, MFM takes advantage of symbolic representation and reasoning, which have enabled a wide range of applications. MFM generally works well in plant-wide operation decision support, such as alarm analysis [15], accident management [16], and risk monitoring [17], which benefit from high-level modeling preference. However, this focus may not be sufficient if the mechanical functionality is also required to perceive how the normal performance of equipment is fulfilled and how the maintenance task can be accurately located on the problematic parts. Although functional modeling has been widely recognized in mechanical design [18], the relevant methods usually do not support maintenance decisions. On the contrary, MFM can integrate sensor data for diagnostics and prognostics [19], but modeling at the mechanical level is required. Therefore, in this paper, we propose a mechanical functional modeling approach based on MFM that is, in principle, distinct from the existing modeling practices using MFM. The proposed approach enables diagnosis and prognosis of mechanical failures based on the monitoring of equipment health conditions. In addition, because MFM uses an unified language across levels of abstraction, e.g., from the mechanical level to the plant level, it can offer reciprocal decision support between maintenance and operation [20], which cannot be realized by the other CBM methods. Hence, in this paper, we also propose a framework of CBM that is able to support

maintenance decisions from the operational perspective, taking advantage of the recently developed MFM-based RCM automation system [13].

The remaining part of this paper is organized as follows: the immediately following section presents the state of the art related the topic in this paper, and the important relationship between maintenance and functional modeling is highlighted. Existing classifications of mechanical functions and their modeling methods are also reviewed. Section 3 proposes a mechanical functional modeling approach. How the approach is coupled with existing process functional modeling to offer the basis of the maintenance support method presented in this paper is also explained. Section 4 proposes a framework of an optimized CBM based on functional modeling coupling and causal reasoning. Section 5 provides a case study based on a test apparatus. Discussion and Conclusions are summarized in the final two sections.

## 2. State of the Art

This section presents the state of the art, highlighting the relationship between functional modeling and maintenance. There are two aspects, both of which contribute to the proposed approach of maintenance decision support. One is related to the determination of maintenance requirements, which is aligned with the RCM method. Another concerns how to develop a knowledge-based diagnostic and prognostic system for maintenance. This section also includes a discussion of existing taxonomies of mechanical functions and how functions can be modeled, both of which are highly relevant to the proposed approach.

### 2.1. Determining Maintenance Requirements

Most industrial facilities adopt RCM to determine their maintenance requirements with respect to what type of maintenance is appropriate for each of their assets. In RCM, the goal of maintenance is to preserve system function rather than to protect the asset itself. It emphasizes the importance of the purpose or the role that equipment can actually serve in operation. As a rule of thumb, equipment whose failure has significant effect on system functions requires preventive maintenance, such as CBM, while for equipment that is less important to system functions, corrective maintenance after failure is acceptable [21].

Functional modeling is recognized as the key process to identify all system functions to be preserved by maintenance. A functional block diagram (FBD) describing the major top-level functions and their interactions is traditionally applied in RCM. However, FBD only defines function as the physical output of system but overlooks the intentions of the system. This inevitably results in the absence of important system functions that need to be preserved. Moreover, FBD focuses on defining functions at the system level, leaving failure consequence analysis, which is required by RCM to identify the relations between equipment failures and system failures, to analyst experience. In a previous study, we developed an AI solution for RCM that can be used to automatically determine the maintenance requirements for massive assets [13]. This RCM automation system was established based on the comprehensive representation of functional relations from the equipment level to the system level.

### 2.2. Knowledge-Based Diagnostics and Prognostics

Given that a standalone piece of equipment can perform at least one significant function, maintenance can be seen as a direct response to the failure of this function, i.e., functional failure, either after or before the failure. An important assumption about CBM is the P–F curve [22], as shown in Figure 1. It is assumed that the condition of equipment can deteriorate in use. Before it reaches functional failure, i.e., point "F", there is usually a potential failure, i.e., point "P", that can be detected. The interval between P and F can be considered as a time window during which maintenance can be performed to prevent the occurrence of functional failure. Therefore, diagnosis of potential failure and prediction of how it could lead to functional failure is the key to determining the perfect timing of maintenance.

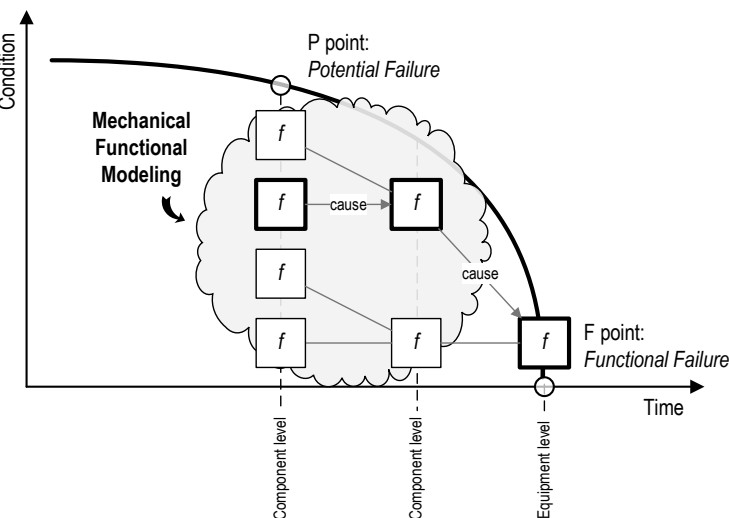

**Figure 1.** P–F curve and mechanical functional modeling.

Many methods have been developed for CBM in order to determine the P point and predict the distance to the F point, using either available math models, data, or knowledge [4,23]. With respect to AI techniques, methods can be divided into those specifically using data, i.e., machine learning, and those specifically using knowledge, i.e., knowledge-based approaches [8,24]. Although data-driven machine learning can produce relatively accurate results, the knowledge-based method usually provides an alternative in cases in which data are lacking for model training. However, establishing knowledge in the form of rules as adopted by traditionally expert systems may be faced with the problem of combinatorial explosion, making it difficult to address new situations [1]. It is therefore suggested to combine rules with symbolic representations for model and rule-based reasoning [16,25].

We argue that the process leading from "P" to "F" is based on a kind of system knowledge, i.e., the functional structure of equipment, considering most equipment develops functional failures, usually as a result of some functional losses its mechanical components. In other words, failure considered at the equipment level or higher can be caused component-level failures. Figure 1 also shows this relationship, demonstrating how failure is propagated across different levels of abstraction within the boundary of equipment. This kind of functional knowledge can provide information that is difficult to express through math or data, which is beneficial for the development of a model-based diagnosis and prognosis system [26]. Many works have attempted to identify or propagate failures using function models [27–29]. However, none of the proposed methods provides maintenance implications, in accordance with Figure 1. The main reason is that they are mostly concerned with the conceptual design of complex systems rather than specific equipment. Accordingly, the modeling is centered around relatively higher levels, which cannot reflect potential failures of equipment. Moreover, all of these function-based failure analyses use FBD as their primary functional modeling methodology, which directly maps according to the physical structure and therefore cannot capture all the necessary functional knowledge. Instead, MFM has a robust capability to represent comprehensive functional knowledge, including both intentions and causality, but a new framework for mechanical functional modeling is required to enable the analysis of mechanical failures.

In this paper, the abovementioned aspects relating functional modeling to maintenance are integrated to develop a maintenance support system. First, the potential of MFM at the mechanical level is explored, which is the key to accommodating condition monitoring data for failure diagnosis and prognosis. Secondly, a framework combining RCM and CBM for cost-effective maintenance decisions is proposed, which uses MFM as the unified knowledge base.

### 2.3. Classification of Mechanical Functions

Ontology is the heart of knowledge representation for any domain and also important for knowledge sharing in that domain [30]. Many researchers have explored the mechanical functional modeling ontology and attempted to establish a classification of mechanical functions. Below are several typical mechanical function classifications presented in the literature. Figure 2 summarizes these classifications and indicates their relationships. For each classification, it is difficult to cover all potential uses because of the wide diversity of mechanical products. The purpose here is to compare different classifications and identify the relevance in representing elementary mechanical functions.

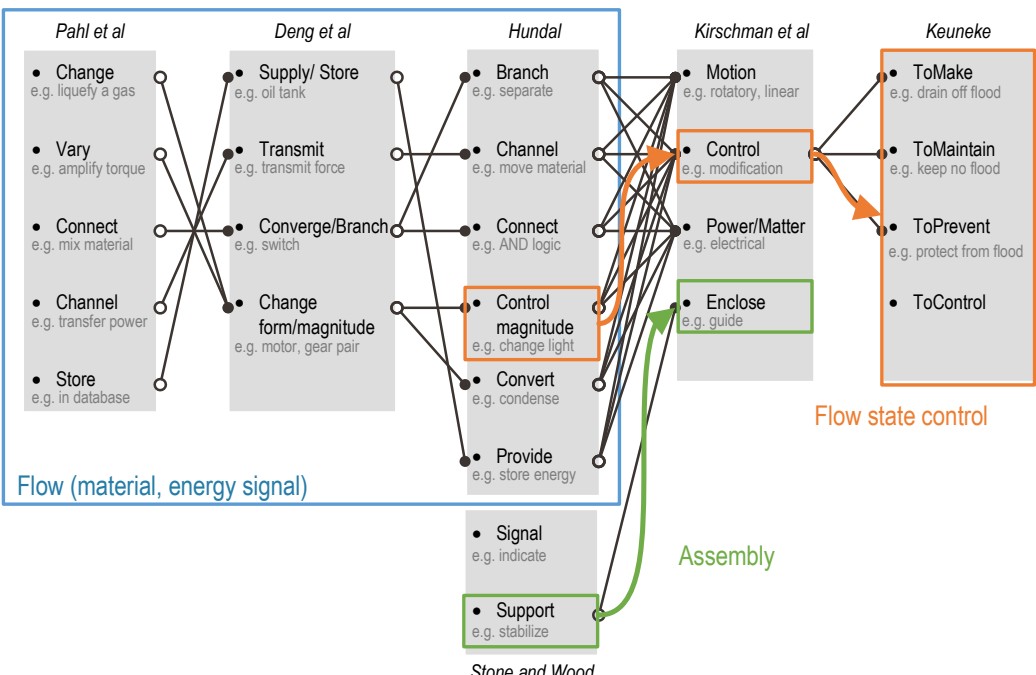

**Figure 2.** Comparison and relationship between mechanical function classification of Pahl et al. [31], Deng et al. [32], Hundal [33], Stone and Wood [34], Kirschman et al. [35], and Keuneke [36].

1. Pahl et al. [31] developed five high-level functions, i.e., generally valid subfunctions, which are defined according to the relationships between inputs and outputs regarding the change in type, magnitude, number, place, and time. The corresponding functions are change, vary, connect, channel, and store.
2. Deng et al. [32]'s classification is derived from work reported by Pahl et al. [31] but uses different vocabulary for the formulation. Mechanical functions are classified into four categories according to whether they are related to supply/store, transmit, converge/branch, or change form/magnitude of material or energy.
3. Hundal [33] broke down the classifications proposed by Pahl et al. [31] and Deng et al. [32] and generated six mechanical function classes that can address flows of material, energy, and signal. On this basis, Stone and Wood [34] also included signal and support. Note that signal here only refers to how a status signal is manifested or used, apart from which control signals can also be processed on the basis of six basic functions. Support means that the signal does not satisfy the functionality required by the product but provides support for the other mechanical functions.
4. Kirschman et al. [35] did not specify the verb form of mechanical functions but focused more on the object form. Four templates for defining mechanical functions were proposed, which are related to motion, power/matter, control, or enclosure.
5. Keuneke [36] did not make a distinction from the viewpoint of material, energy, or signal flow, instead focusing on the state change. There are four function types, i.e.,

ToMake, ToMaintain, ToPrevent, and ToControl, among which ToControl has the power to execute the other three functions.

Whatever a technical system is, either plant or machine, it is always involved in processes of channeling or converting material, energy, and signals, which can be considered flows from inputs to outputs [31]. Most flow-term mechanical function classifications, e.g., that proposed by Deng et al. [32], Hundal [33], and Stone and Wood [34], have their root in Pahl classification [31], which almost covers all possibilities in terms of how a flow can be processed. Kirschman et al. [35] classified mechanical functions not in terms of the action of the function but according to the object in which motion is enabled by force and, ultimately, can be explained in terms of energy. Thus, like for power/matter, motion can also be represented as flow. Note that energy flow usually has variant forms, which can also involve the flow of forces, torques, etc. [31]. Enclose in [35] is similar to support as described in [34], not processing material or energy on its own but providing the condition that enables the material or energy functions. Flow-term functions can also be subject to control [35] to change the states of functions, which may take several other forms [36]. In summary, when considering the representation of elementary mechanical functions of assembly parts of equipment, it is necessary to develop a modeling approach that includes the following points:

- Functions of various flows as defined by Pahl et al. [31];
- Conditions for flow functions;
- Control functions if control equipment is involved.

### 2.4. Literature Review for Functional Modeling Methods

The purpose of functional modeling is to represent the designer's intentions [18], which involve not only how an elementary function such as that presented in Section 2.3 is described but also a description of how a function is accomplished by its physical structure and how functions interact with one another [37]. Regarding how an elementary function can be represented, there are three categories of methods [38],

1. Verb–noun pairs are how people imply function in ordinary language.
2. Input–output (I–O) flow transformation defines a function as a relation between input and output of flow, which is relevant to the flow-term functions shown in Figure 2. Transformation from input flow to output flow can be recognized as a task purpose, i.e., function.
3. Input–output (I–O) state transformation characterizes function as a sequential change from the initial state to the ending state, as state change usually occurs when the function is being delivered.

As a form of natural language syntax, verb–noun pairs can be used as the basic form to describe the other two representations [39]. For instance, the function of a heat exchanger is to"reduce heat", which, in the expression of flow transformation, is the relation between the inlet energy and a certain outlet energy. The same function can be understood as the state transformation as the temperature state changes from inlet to outlet. However, without reference to the intentional knowledge of humans, both flow transformation and state transformation are merely representations of behavior. Functional modeling requires the establishment of links between human needs in the subjective realm and physical behaviors in the objective realm [18]. Pahl et al. [31] and Umeda and Tomiyama [40] combined flow transformation with the decomposition of human needs so that an abstract function could be decomposed into subfunctions, which are then associated with their own inputs and outputs. Deng [39] also proposed a function-decomposition-mapping model to include both functions and non-functions.

Some other researchers have considered relationships with structure, in addition to using functions to bridge the gap between needs and behaviors, since any usage of a functional model is related to decisions at the structural level, regardless of the design or operation. Gero [41] developed a function (F)–behavior (B)–structure (S) model as a design

prototype, which is considered a transformation from intentions to the structure. Similarly, the function (F)–environment (E)–behavior (B)–structure(S) model [39] specifies a set of physical structures necessary to achieve functions. Chandrasekaran and Josephson [42] suggested decomposing device-centric functions into subfunctions so that they can be associated with physical phenomena and components of the design object. Lind [14] proposed the framework of goal (G)–function (F)–component(C), which motivated the development of MFM. The purpose of the G-F-C framework or MFM is to help the operator of a complex system cope with complexity. The unique aspect of the proposed model is that it describes the means–end relations, i.e., intentional knowledge between the goal, function, and component, without losing structural information, which is represented by the part–whole decomposition. This is extremely important for operation decision support because the operator uses the function concept to understand how the system works but ultimately operates according to the structure.

## 3. Mechanical Functional Modeling

This section introduces a mechanical functional modeling approach by applying MFM.

### 3.1. MFM

MFM offers a formalized language for functional modeling, which expresses the potential of representing both elementary mechanical functions and the mechanical functional structure of equipment. Figure 3 shows the framework and symbols of MFM. The purpose of MFM was originally to decompose a complex system to a level that can easily be mentally processed by human beings. Decomposition is achieved through two dimensions, i.e., means–end and part–whole. First, MFM is completely goal-oriented. Depending on the chosen goal, a system can be explained from many perspectives in terms of a means–end structure, from goal to function and the physical component. Note that in many cases, the goal or function can provide a means for the achievement of other functions contributing to a different goal. For example, Figure 3 shows how means–end structure 2/3 connects to 1 and means–end structure 4 connects to 2. On the other dimension, each means–end structure can be aggregated along the part–whole relation, e.g., means–end structure 2 connects to 3. Multiple goals can be aggregated as a super goal, and functions and components can also be integrated [14]. In comparison with FBD, the merit of MFM is that it does not simply connect functions of individual components; rather, it describes a kind of behavioral interaction between components.

MFM has sufficient semantics and syntax to represent functions and their relations, as shown by various types of modeling symbols in Figure 3. MFM adopts the flow term to represent mass/energy flow and signal/control flow. But unlike the conventional flow transformation of function [31], which is essentially a black-box module displaying only inputs and outputs, MFM can elaborate what happens between sinks and sources as mass flow structure or energy flow structure. Apart from describing function in terms of flow transformation, MFM can also describe function as state transformation. In MFM, several qualitative states are defined for all six types of flow function in terms of time rather than place. Each function can alter its state either by itself through action or through an outside influence. MFM has been updated with sophisticated cause–effect rules in specific function–relation patterns, which enhance the causal reasoning capability. The rule-based causal reasoning of MFM is the core technique in diagnostics and prognostics for maintenance, as elaborated later.

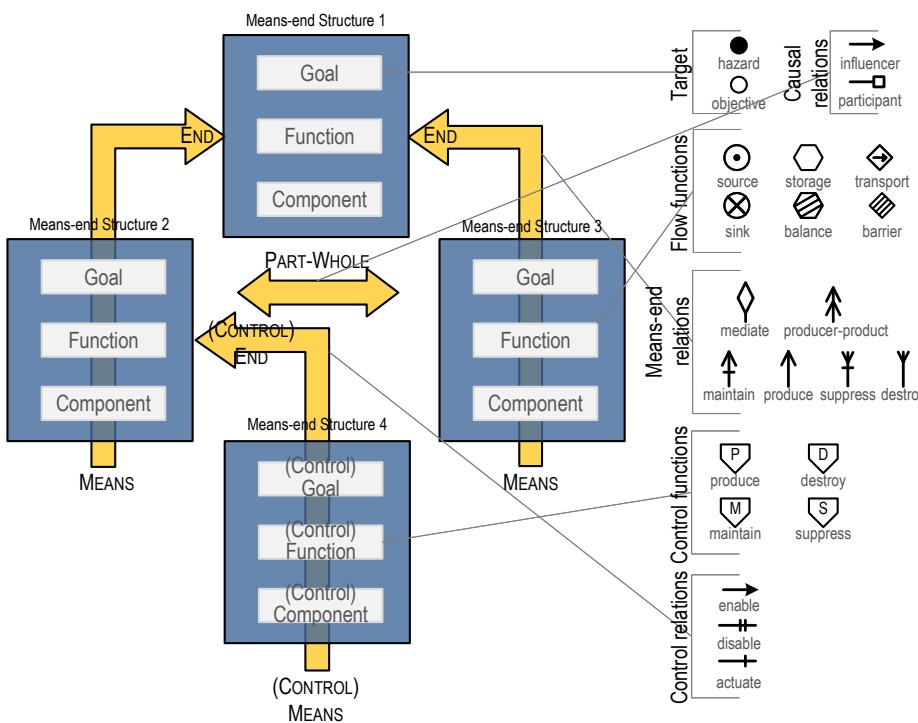

**Figure 3.** Introduction to MFM: framework and symbols.

### 3.2. Problem with the Process-Oriented Approach

Traditionally, the modeling procedure for MFM is technically process-oriented, where "process" means an industrial process that can involve chemical, physical, or electrical processes to aid in the manufacturing of products, and is usually carried out on a very large scale [43]. This is because most existing applications of MFM are performed in process and control engineering for operation decision support. In order to build an MFM model for a process plant, stream analysis and objective tree analysis are two important steps [44]. Stream analysis is used to capture all mass and energy flows involved in the working materials being processed, such as water or heat. This task is usually performed by following a process flow diagram, such as a piping and instrumentation diagram (P&ID). Then, different identified streams can be integrated in accordance with an objective tree, which is a hierarchical structure showing how different purposes of the system are arranged along the means–end dimension. Figure 4 shows how a central heating system [45] can be modeled by applying the process-oriented approach. First, three streams can be identified for the system, which are the energy flow from burner to radiator, water cycle, and energy conversion in the burner. These streams, which are represented as flow structures, have their own goals, which comprise the objective tree of the system. The tree can guide the modeled flow structures to be connected. For example, water cycle can be seen as the means of transforming energy via water.

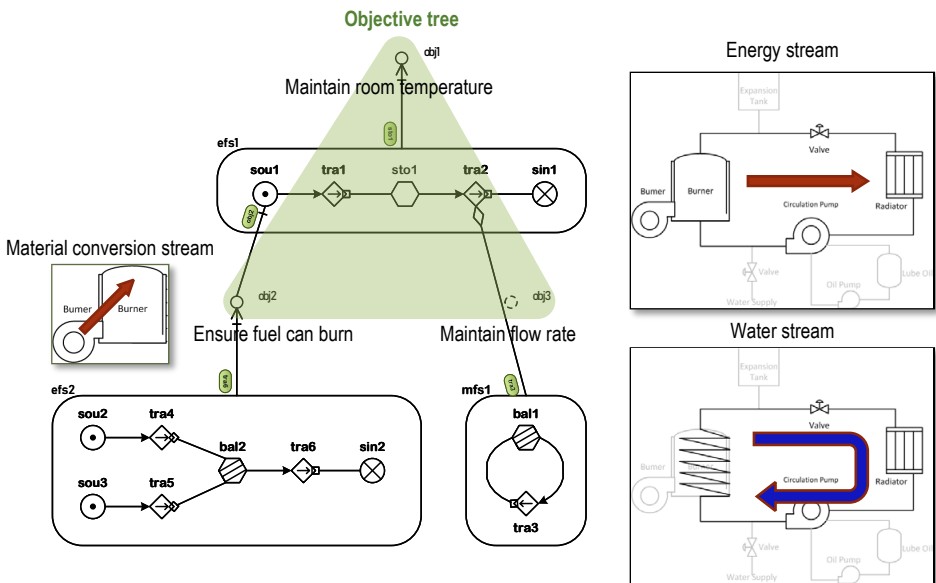

**Figure 4.** Functional modeling for a central heating system using the traditional process-oriented MFM approach.

However, mechanical functional modeling is fundamentally different from functional modeling in process engineering. Mechanical functional modeling is usually concerned with a small piece of a process plant rather than the plant at a large scale, for, example the pump in Figure 4. Accordingly, in the function model, mechanical function modeling should focus on how the function of the pump "transporting water", i.e., *tra*3, is achieved by interactions between various mechanical functions inside the pump. It should be noted that what contribute to the achievement of mechanical functions are the physical interactions between different mechanical components rather through transmission of material and energy of working products. In other words, a mechanical system exhibits certain functions or behaviors through the interactions that occur between its mechanical components, which are mainly governed by physics laws [39]. Pahl et al. [31] argued that with regard to physical interactions, the concept of force is essential, which should be conceived as the means by which the motion of mass is changed. This process can be further explained as the change in mechanical energy. In addition to mechanical energy, interactions include transfers of the other types of energy, such as thermal and electrical energy; materials such as liquids and solids; and signals like control impulse signals. Therefore, flow is still the main manifest form of mechanical functions and can also be found in many classifications of mechanical functions. Even then, given the diversity of physical interactions and the lack of obvious contiguity of flow interactions between components, it is relatively difficult to perform a stream analysis of mechanical equipment as a whole by simply following mechanical drawings. It is more suitable to analyze the input–output relationships of individual mechanical components of equipment that can form flows; then, functional integration is possible.

### 3.3. Component-Oriented Functional Modeling

We therefore propose a component-oriented approach to address mechanical functional modeling based on the decomposition of mechanical components. Figure 5 shows the process of the proposed mechanical functional modeling, which can be used to convert functional knowledge of a mechanical piece of equipment into a function model that can be applied to diagnosis and prognosis of mechanical failures.

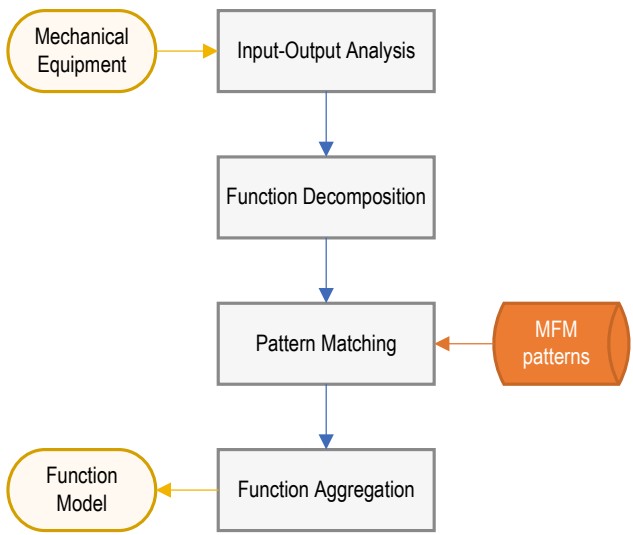

**Figure 5.** The process of mechanical functional modeling.

### 3.3.1. Function Decomposition

Every component of a piece of equipment is connected to its context by means of inputs and outputs [31]. As shown in Figure 6, consider a mechanical item that is composed of different components or parts, each of which has several input or output interfaces for interchange of different types of object. According to the discussion presented in Section 2.3, the objects of flow include mass, energy, signal, and motion. Motion flow is usually associated with force flow, which is further related to either transfer of energy (energy flow) or movement of mass (mass flow). Note that force flow may not transfer any energy or mass but only provide the assembly condition that can support functions of equipment. Therefore, six types of flow object that are included in the proposed mechanical functional modeling, that is, (1) mass, (2) energy, (3) signal, (4) energy-related force, (5) mass-related force, and (6) assembly-related force.

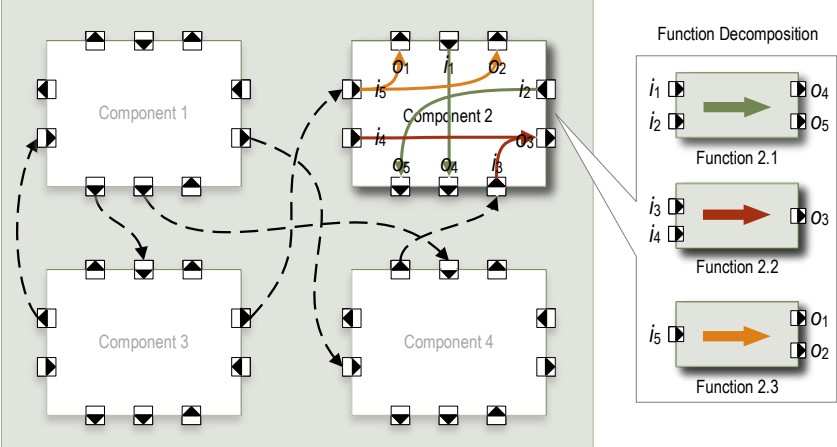

**Figure 6.** Component-oriented functional modeling: from input–output analysis to function decomposition.

By using input–output analysis, each component in a piece of equipment can be decomposed into distinct functions, each of which is represented in terms of flow transformation based on a specific combination of inputs to that of outputs. According to Zhao et al. [46], an input/output-like function ($f_j$) can be represented as a binary function, i.e., $< I_j, O_j >$, in which $I_j$ represents a set of functional flows at the input ports, i.e., $\{i_k\}$ ($k = 1, 2, \ldots, m$), where $i_k$ is the $k$th input flow and $m$ is the number of input flows; while $O_j$ represents a

set of flows at the output ports, i.e., $\{o_t\}$ ($t = 1, 2, \ldots, n$). $i_t$ is the $t$th input flow, and $n$ is the number of output flows. Therefore, the mechanical function(s) of component $c$ can be expressed as:

$$F_c = \{f_j\}(j = 1, 2, \ldots, s) \tag{1}$$

where $s$ denotes the number of functions. For instance, in Figure 6, component 2 is decomposed into three mechanical functions depending on the chosen combination of inputs and outputs, i.e., $\{< \{i_1, i_2\}, \{o_4, o_5\} >, < \{i_3, i_4\}, \{o_3\} >, \text{and} < \{i_5\}, \{o_1, o_2\} >\}$. Objects may be transmitted from one component to another, as shown by the dashed arrows, which implies that mechanical functions of different components can interact with one another. However, it should be noted that functions can also interact with one another without any transmission of an object. This requires further analysis of means–end relations, which will be explained later.

### 3.3.2. Mechanical Function Patterns

It is necessary to know what combination of inputs and outputs can represent an exact mechanical function. As mentioned before, Pahl et al. [31] provided a sufficient definition of flow-type mechanical functions. Accordingly, mechanical function patterns were developed using the semantics of MFM, as shown in Figure 7. The difference between each pattern lies in how the flow from inputs to outputs can be processed inside the black box. Patterns are represented in terms of function structure in MFM using symbols that can constitute flows. The number of inputs and outputs being modeled as source and sink, respectively, can vary from case to case.

- The function of change ($F_{chg}$) can be expressed as flow, indicating that the form of the flow object being processed has been converted from inputs to outputs. But it should be noted that the objective type—either mass or energy—does not change, which allows inputs and outputs to be included in the same mass flow structure or energy flow structure. The pattern shown in Figure 7 consists of several source–transport that represent input flows before their types change and several pairs transport–sink pairs the represent output flows after the change. A balance function is used to connect inputs and outputs.

$$F_{chg} =< \{i_k\}, \{o_t\} > (i = 1, 2, \ldots, m; t = 1, 2, \ldots, n) \tag{2}$$

- The connect function ($F_{con}$) expressed below can have more than one input and output, but the flow object does not change its type or form. The *converge* ($F_{cog}$) and *branch* ($F_{brh}$) functions are also defined within this pattern depending on the number of input flows and output flows. The pattern $F_{con}$ shown in Figure 7 has the same form as $F_{chg}$. The difference is that neither the type nor form changes from input flows to output flows.

$$F_{con} =< \{i_k\}, \{o_t\} >= \begin{cases} F_{con}, & m = n \\ F_{cog}, & m > n, (i = 1, 2, \ldots, m; t = 1, 2, \ldots, n) \\ F_{brh}, & m < n \end{cases} \tag{3}$$

- The *channel* function ($F_{chn}$) is a flow transmitting objects from only one input to only one output, during which time the object does not change its type or form. A *balance* is used to connect upstream and downstream flows.

$$F_{chn} =< i_1, o_1 > \tag{4}$$

- The *store* function ($F_{str}$) indicates object accumulation, which can be switched to *Supply* ($F_{spl}$) if there is no input. However, it does not involve a change in object type or form.

$$F_{str} =< \{i_k\}, \{o_t\} >= \begin{cases} F_{str}, & m \neq 0 \\ F_{spl}, & m = 0 \end{cases}, (i = 1, 2, \ldots, m; t = 1, 2, \ldots, n) \qquad (5)$$

The vary function, i.e., change in magnitude, is not defined by a specific pattern in terms of flow from input to output. Rather, vary is a function that can affect the other four types of mechanical function by changing their state in a qualitative way. As shown in Figure 7, the vary function is realized by changing the state of a specific flow function, e.g., transport. For example, for the mechanical pattern of the channel function (see Figure 7), when *tra-I-1* increases, *tra-O-1* increases according the causal feature of MFM, which implies that the capability of channel increases. An important aspect of vary is that it can be associated with condition monitoring data, changes in which can be reflected in the corresponding mechanical function. In other words, the conditions of mechanical functions can be confirmed by monitoring whether there is a vary function that is acted upon, which can reflect either degradation of components or failure propagation from the other components.

The above patterns are designed to represent mass- or energy-flow mechanical functions. Force-flow mechanical functions can be represented represented as patterns in mass or an energy form depending on which the function is related to. Below, an example of a rotor is used to illustrate how a mechanical component can be decomposed into different fundamental mechanical functions and explicitly represented using defined modeling patterns. As shown in Figure 8, it is assumed that the rotor is driven by a motor. It functions normally by receiving support only from two directions, i.e., upper and lower bearings.

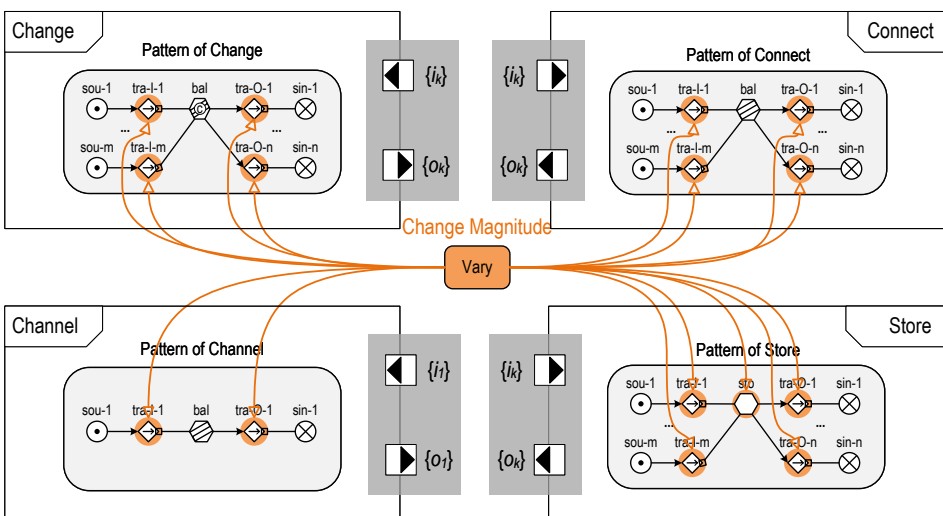

**Figure 7.** Definition of mechanical function patterns that can be used to represent the decomposed functions. The patterns for change, connect, channel, and store are defined in the form of input–output flows. The vary function is not defined according to a pattern but can change the magnitude of inputs or outputs of the other four categories of mechanical function.

The input–output analysis shows that the rotor is able to transmit different flows, among which the force flow is the most essential because it can explain both motion of mass and transfer of energy. As shown in Figure 8, there are four force flows, each of which can be used to further derive mass flow or energy flow. The rotor can be decomposed into various elementary mechanical functions and represented as flow structures using the defined modeling patterns.

### 3.3.3. Relations between Mechanical Functions

Different mechanical functions should be interacted with according to relations. Chandrasekaran [47] indicated that a causal link between two state transitions can be explained in four ways by appealing to:

- `By-CPD`, i.e., Causal process description, which is the causal knowledge that people compile through problem-solving experience, making it goal-dependent. It is usually couched in terms of common sense or scientific ontology. For example, closing a switch causes voltage to be applied between terminals.
- `By-Function-of<>`, where the blank can be filled with a device component, which means that it is the function of the component that causes the transition.
- `By-Domain-Law`, which corresponds to mathematical equations that embody scientific laws and express relations between state variables. For example, according to Ohm's law, voltage 'causes' a current.
- `By-Qualifier`, which corresponds to a conditional relation that is required for the transition to take place. This usually happens for a support function, where a state, e.g., support force, must hold for another state transition.

All above explanations of the causal links of state transitions can be adapted to MFM to represent interactions between the decomposed mechanical functions. We should also be emphasize the distinction between function interactions inside the same component and those across different components. Chandrasekaran and Josephson [42] viewed this difference, essentially, as a distinction between behaviors within the artifact boundary of the design, i.e., device-centric, and purposed that the context of an artifact is required for justification of its fulfillment, i.e., environment-centric. The device-centric view asserts the intrinsic functional hierarchy of the component. For example, transmission of torque can bring about transmission of rotational energy, both of which occur in the rotor. On the other hand, the environment-centric perspective implies another kind of interaction, i.e., that functions of different components can interact with one another using context to achieve a goal specified by the user. For instance, transmission of rotational energy can ultimately drive the impellers of a pump, utilizing rotational energy to transport water. Transporting water may not be the intrinsic purpose of a rotor, but here it is considered as such in the context of a pump. Table 1 lists the relations in MFM that can appropriately describe the causal links between mechanical functions. With suitable relations, the discrete functions shown in Figure 8 can be connected, as shown in Figure 9. Because only one component, i.e., the rotor, is modeled, only device-centric functions are considered, for example, how the laws of motion govern the vertical movement of the rotor.

**Table 1.** Relation types for establishment of interactions between mechanical functions.

| Causal Link | MFM Relation * | Type | Scope |
|---|---|---|---|
| `By-CPD` | ⟶≫ | Device-centric Environment-centric | Knowledge about goal achievement that one function can 'cause' another. |
| `By-Function-of<>` | •——• | Environment-centric | Functions of different components interact with one another by providing/receiving the same flow. |
| `By-Domain-Law` | —◇ | Device-centric Environment-centric | The causal relations between variables are governed by domain laws. |
| `By-Qualifier` | ——+ | Environment-centric | An objective achieved by one function can enable another. |
| | ——#— | Environment-centric | An objective achieved by one function can disable another. |

\* See Figure 3.

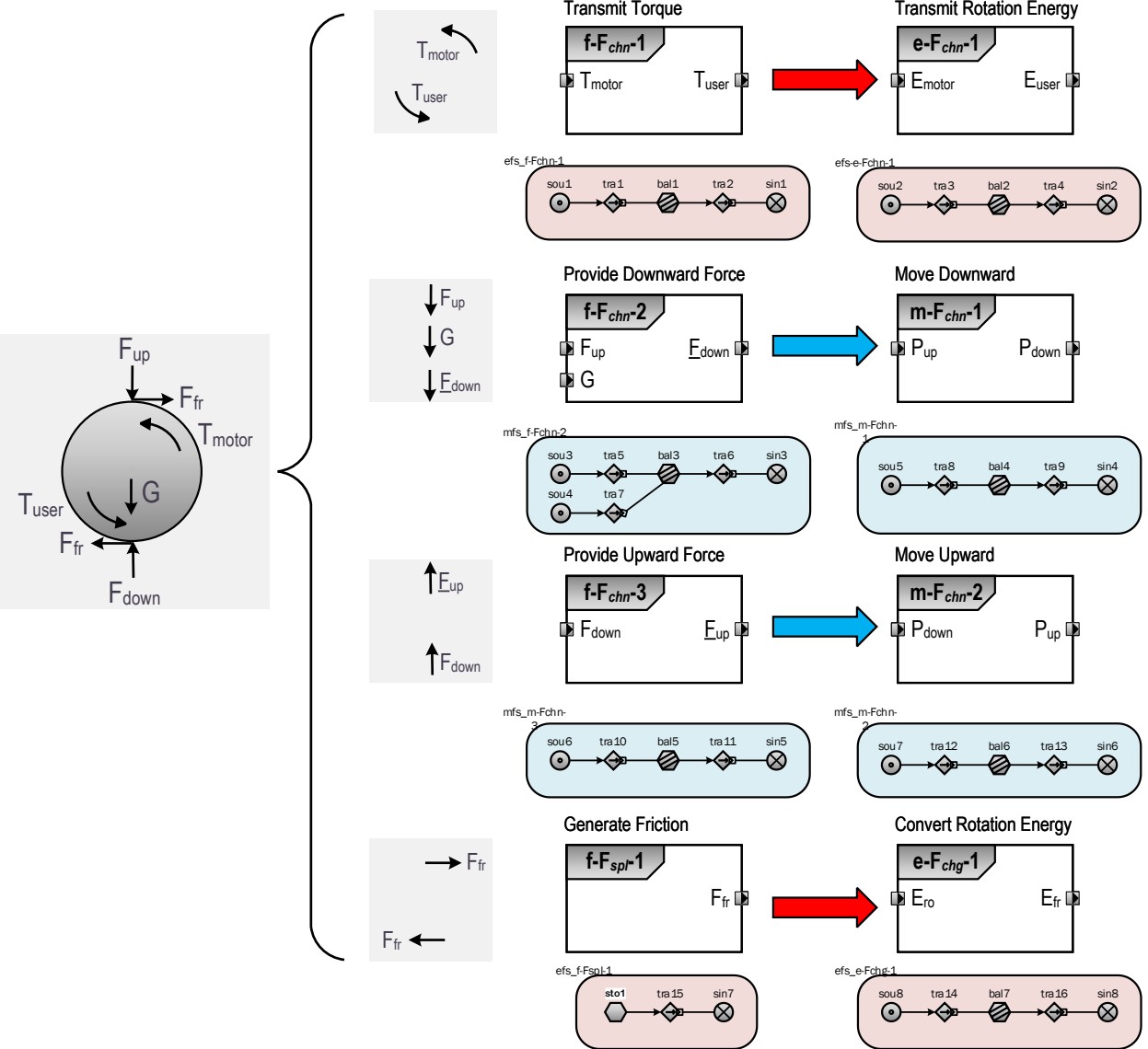

**Figure 8.** Function decomposition of a rotor. ($T_{motor}$: torque received from the motor; $E_{motor}$: rotation energy received from the motor; $T_{user}$: torque transferred to the user; $E_{user}$: rotation energy transferred to the user; $F_{up}$: force received from the upper bearing; $G$: gravity; $\underline{F}_{down}$: force transferred to the lower bearing; $P_{up}$: position of the upper side of the rotor; $\underline{F}_{up}$: force transferred to the upper bearing; $F_{down}$: force received from the lower bearing; $P_{down}$: position of the lower side of the rotor; $F_{fr}$: generated friction; $E_{ro}$: rotation energy; $P_{up}$: energy loss due to friction. Rule of numbering a mechanical function: $([a] - F_{[b]} - [c])$, where $a$ indicates whether the function is a mass flow ($a = m$) or energy flow ($a = e$); $b$ indicates the type of pattern required to represent the function, as defined in this section; and c is the functions, which is countered based on the pattern type).

### 3.4. Coupling between Mechanical Functional Modeling and Process Functional Modeling

Owing to the use of an identical modeling language, i.e., MFM, the component-oriented mechanical functional modeling introduced in Section 3.3 can be coupled with modeling of high-level functions of industrial plants by adopting a process-oriented approach. This coupling is the key of the proposed maintenance decision support system. Figure 10 shows how the mechanical functional modeling of a pump is integrated into the process functional modeling of a nuclear power plant. The process-oriented method combines stream analysis and objective tree analysis to construct a function model. The former is used to capture all mass and energy flows and their interactions involved in the working products being processed, such as cooling water. The latter is used to decompose

streams into a hierarchical description [44]. As shown in Figure 10, at the process level, the function of the pump is manifested as pumping. This function can be seen as a medium that connects the functional modeling of two different realms. At the mechanical level, the "pumping" function can be further decomposed to show how it can relate to mechanical functions. An important implication of modeling coupling for the maintenance decision is that failures of functions at the mechanical level can propagate to functions at the process level, which may result in operational problems.

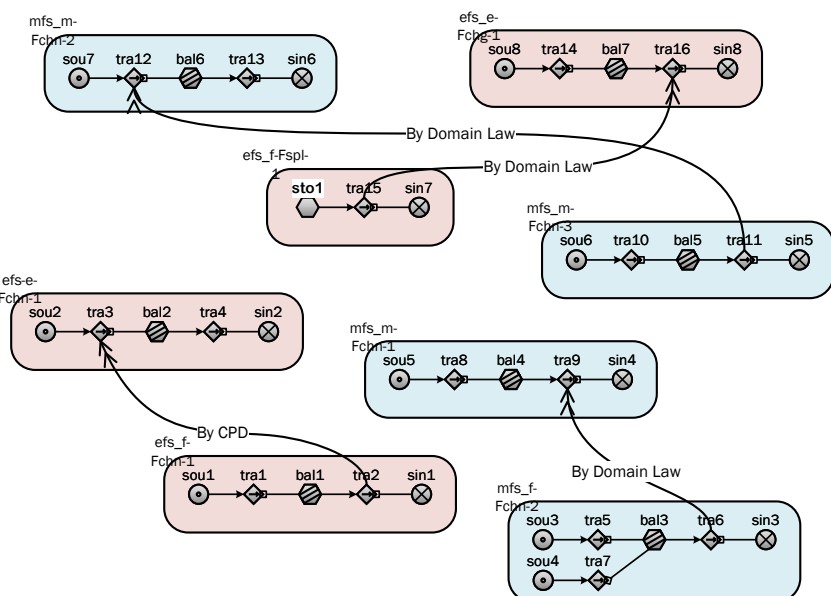

**Figure 9.** Function aggregation between mechanical functions.

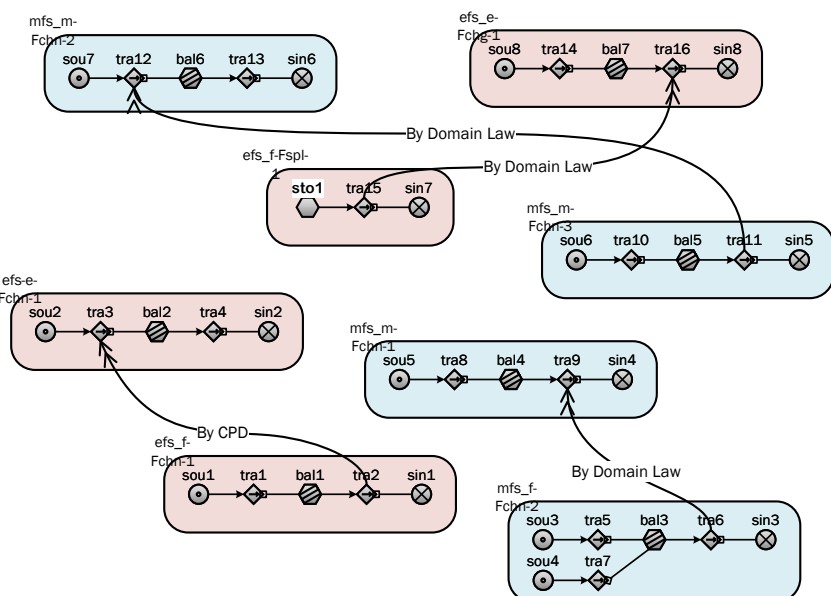

**Figure 10.** Coupling between component-oriented mechanical functional modeling and process-oriented functional modeling for a process plant.

## 4. A Framework of Optimized CBM Combined with RCM

In this section, a framework of optimized CBM combined with RCM is proposed, which takes advantage of the mechanical functional modeling approach proposed above,

as well as its coupling with process functional modeling. The previously developed RCM automation system [13] is briefly introduced below. Because the causal reasoning capability plays the key role in the proposed maintenance decision support, it is also explained.

A good maintenance strategy must satisfy two criteria, i.e., cost-effectiveness and accuracy. Although CBM is regarded as the superior maintenance method capable of ensuring maintenance accuracy, its cost is also significant. In other words, not all assets in a large-scale facility deserve CBM. RCM can help to effectively organize CBM in optimizing the maintenance solution by determining the importance of equipment in order to justify investment in CBM [48]. Therefore, a combination of CBM and RCM is a suitable means to ensure that maintenance is both accurate and cost-effective. In our proposed maintenance decision support framework, this is realized by diagnostic and prognostic reasoning using MFM at both mechanical and process levels. Figure 11 shows the proposed framework of the optimized CBM. All system monitoring data are from the supervisory control and data acquisition (SCADA) system, which includes two types of information: (i) process parameters that are the responsibility of operators during normal plant operation and (ii) equipment health condition states that are the responsibility of maintenance personnel. Before being used for diagnosis and prognosis, those raw data have to be processed to extract features that are required by diagnostic or prognostic algorithms [4]. MFM mainly processes qualitative trends of value data, such as high or low trends, but how the data processing is performed is beyond the scope of this paper. By adopting the inherent causal reasoning capability of MFM, the optimized CBM can offer three kinds of maintenance support, which benefit from the coupling between the process function model (PrFM) and the mechanical function model (MeFM).

Type-1 Maintenance Decision: Diagnostic reasoning PrFM → MeFM identifies mechanical failures that have already occurred (green arrow).

Type-2 Maintenance Decision: Prognostic reasoning MeFM → PrFM via the RCM automation system indicates whether CBM is necessary (red arrow).

Type-3 Maintenance Decision: Diagnostic reasoning MeFM → MeFM identifies potential mechanical failures (blue arrow).

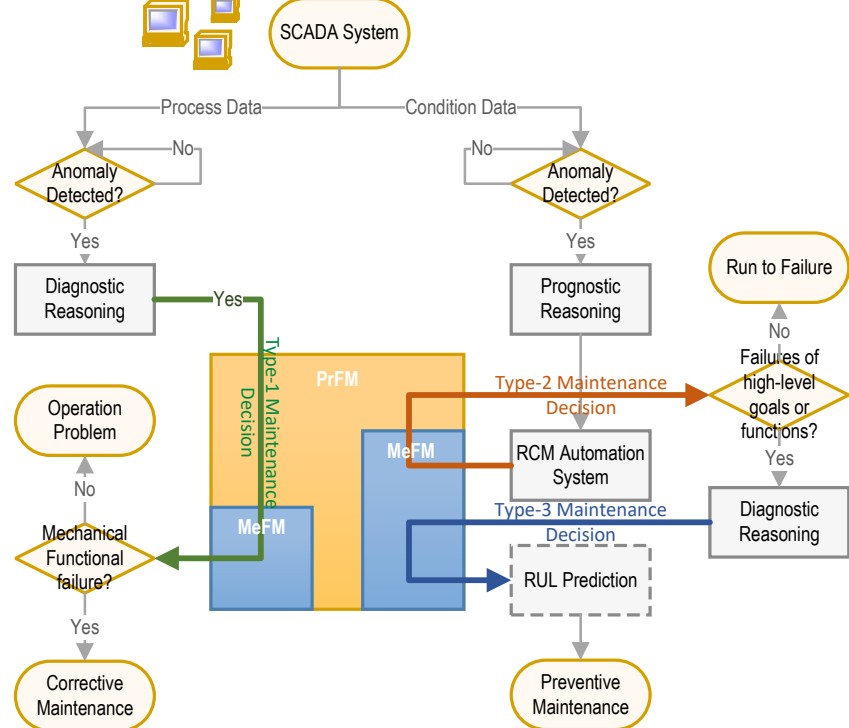

**Figure 11.** Framework of the optimized CBM.

### 4.1. Causal Reasoning of MFM

MFM takes advantage of qualitative modeling applied to condition monitoring, i.e., the fact that qualitative representation is consistent with the natural description and mental model of technical personnel such as designers, operators, and maintainers [49]. The main characteristic of human intelligence is reasoning in terms of ranges of values rather than actual values, which also applied to artificial intelligence [47]. For example, when describing a causal relation between two variables ($x$ and $y$), 'IF $x$ increases, THEN $y$ decreases'. Defining ranges of values is also relevant to changeable features such as deviation, which is useful in defining and reasoning about failures.

MFM defines the trend-form descriptions as cause–consequence patterns, which can be used as causal reasoning rules for both diagnostic and prognostic reasoning. Each function primitive of MFM is specified by a series of discrete function states, such as high or low states. For all possible function pairs in a function model, there are rules that can be expressed as 'CAUSE [function primitive] [function state], CONSEQUENCE [function primitive] [function state]'. Rules in MFM are defined in accordance with first principles such as mass balance and energy balance and are therefore normally independent of a specific application. Therefore, those rules do not lose generality when the functional modeling scope of MFM has been extended from the process level to the mechanical level as proposed in this paper. Figure 12 shows an example of an MFM causal reasoning rule. The full set of rules can be found in [45].

| source/storage (in/pa) | pattern | transport |
|---|---|---|
| low low | | high high |
| low | | high |
| high | | low |
| high high | | low low |

| transport | pattern | (in/pa) sink/storage |
|---|---|---|
| high high | | high high |
| high | | high |
| low | | low |
| low low | | low low |

**Figure 12.** An example of an MFM causal reasoning rule. Red represents causes (source: [45]).

Causal reasoning rules can be used in two directions: to reason about causes for a known situation and to reason about the consequences of an observed situation. Although rules are limited for a short series of causal sequence (i.e., connection between two or three function primitives), the use of rules is also model-based [16]. When multiple rules are continuously applied, e.g., a consequence resulting from a rule may be used as a cause that applies to another rule, it is possible to explain a long series of causal chains within a function model. Considering a function model usually has a complex network of interactions, causal reasoning based on rules generates many possible causes in a diagnostic task or consequences in a prognostic task.

### 4.2. RCM Automation System

In a previous study [13], an RCM system was developed that can be used to automatically determine whether an asset deserves CBM. The system is based on the prognostic reasoning of an MFM model established at the process level. In accordance with the principles of RCM, every functional failure of equipment is evaluated in terms of its consequences for high-level process functions [21]. In the framework of optimized CBM, the developed RCM automation system is linked to the prognostic reasoning from MeFM to PrFM (Type-2 Maintenance Support), which can indicate whether a detected equipment condition anomaly has significant impacts on operation from the perspective of either

safety or availability. This decision informs whether further deep diagnosis of potential failures is needed. Because RCM can also be dynamic [13], in different operational contexts, a condition anomaly may have different consequences in terms of importance, which leads to different maintenance requirements. Applying RCM allows for determination of whether failure detection requires further action. This ability cannot be offered by existing CBM methods.

### 4.3. Diagnostic and Prognostic Reasoning at the Mechanical Level

A function model resulting from the proposed mechanical functional modeling approach was designed for diagnostic and prognostic reasoning about mechanical failures. For this purpose, it is assumed that mechanical functions can be involved in three categories of information space, as shown in Figure 13.

- Operation space (OS) includes the mechanical functions that can accommodate the operation monitoring data, which, in general, measure the overall performance of equipment. Failures of those mechanical functions are considered functional failures.
- Condition space (CS), covers the mechanical functions that can accommodate the sensor data that are related to the health condition of equipment.
- Failure space (FS) does not link to any sensor data but can relate to them through causal relations. Failures of mechanical functions in this area can propagate to functions that are monitored.

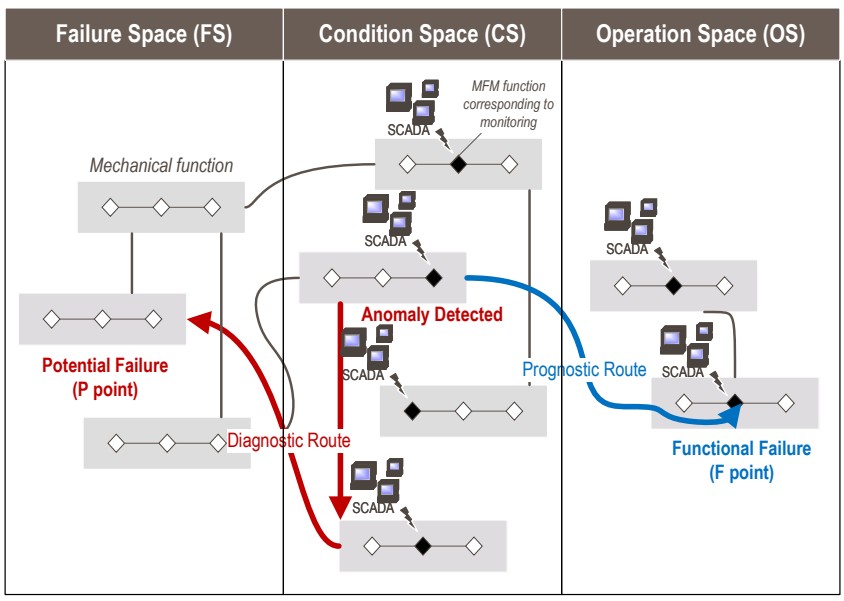

**Figure 13.** Diagnostic and prognostic reasoning at the mechanical level.

For every MFM function primitive that is directly associated with condition monitoring data, when the measured parameter exceeds the prescribed threshold, a qualitative state change (high or low) is transferred to the model to trigger the reasoning process. As mentioned before, depending on whether the direction and use of rules, there are two independent reasoning orientations that can lead to distinct conclusions. One shows the potential consequences as the failure develops. The other searches the possible causes that result in the currently detected abnormal condition. Note that although each reasoning run is based on a single failure, which normally leads to many candidates in FS, combining cause reasoning for different detected condition anomalies may help to validate parts of the diagnostics routes so that the number of potential failures is reduced.

## 5. Case Study

Because the proposed maintenance decision framework involves the coupling of functional modeling at the process and mechanical levels, we present an example combining a process facility and mechanical equipment. As shown in Figure 14, the chosen process plant is an offshore platform for oil production, while mechanical functional modeling is conducted on a test apparatus based on a rotor-lever system. It is assumed that the test apparatus, as a whole, can serve the function of a blower, which is a component critical for the operation of the offshore platform. In other words, the function modelings of the two different levels of abstraction are connected through the overall function of the equipment. In this section, by applying the proposed maintenance decision framework, we illustrate how the detected process anomaly can be used to identify the operation problem, as well as the corrective maintenance insights, and, more importantly, how the detected equipment condition anomaly can be used for determination and implementation of the preventive maintenance strategy. Since the proposed functional modeling approach is designed for mechanical equipment, in the following sections, we elaborate on how a function model for the test apparatus was constructed. The functional modeling of the process plant and how it can be linked to the test apparatus model is only briefly mentioned in our analysis of maintenance decision support.

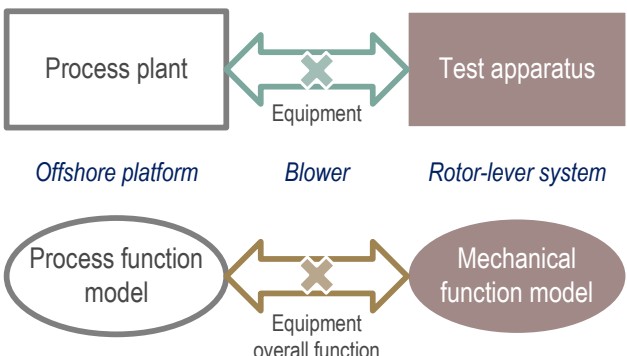

**Figure 14.** Overview of the industrial example described in the case study.

### 5.1. Description of the Test Apparatus

Figure 15 shows a photo of the test apparatus. The core of the test apparatus is a rotor-lever system (lower right of Figure 15), whose schematic is separately shown in Figure 16. The test apparatus is essentially a controllable fluid for film bearings. As shown in Figure 16, the rotor-lever system can be seen as a single rigid body. The rotor is connected to the lever by a pair of conical roller bearings. The whole rotor-lever system is pivoted at the extremity with ball bearings. Forces are applied to the middle point of the lever via a pair of actively lubricated tilting pads. The pair of tilting pads inside of the bearing housing constrains the amplitude of the arm movements to very small angles, which means that small vertical displacements of the rotor center positioned in the middle of the lever are assured. The pair of tilting-pad bearings allow the rotor to only move in the vertical direction. Oil pressure and flow that can provide the fluid film forces applied to the rotor are generated by a hydraulic unit. The fluid lubricant flows from a lubricant tank, then through a filter and up to the high-response servo valve, which is connected to the pair of tilting pads by means of pipelines and is responsible for generating the active fluid film forces on top of the hydrodynamic forces generated by the low-pressure lubricant flow. The injection pressures in the pair of bearing pads are measured by the pressure sensors and can be controlled in order to maintain the vertical displacement of the rotor-lever body.

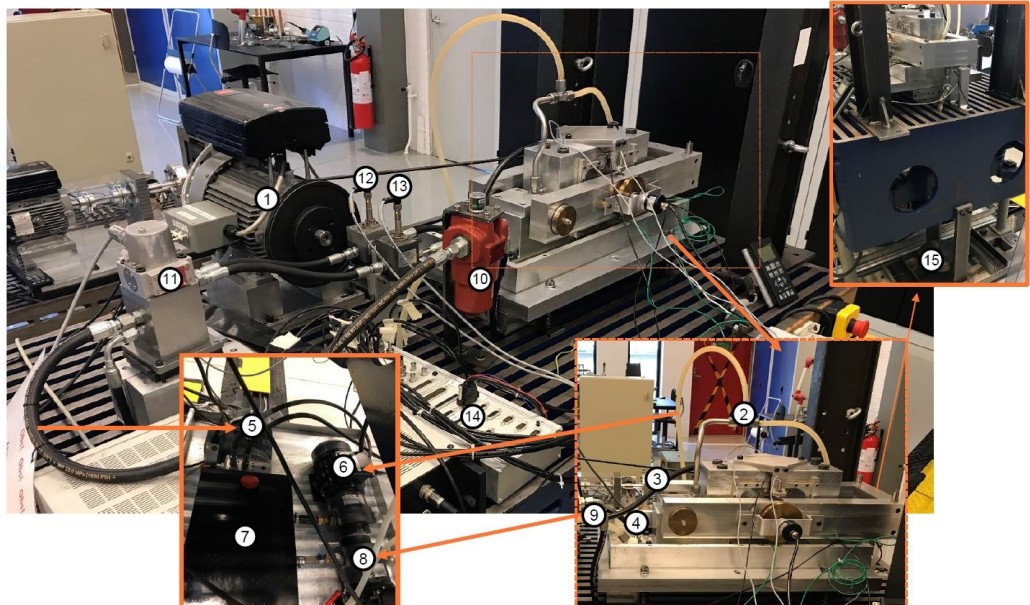

**Figure 15.** Photo of the test apparatus. The components include: an (1) AC motor, (2) low-pressure injection, (3) high-pressure injection (upper), (4) high-pressure injection (lower), (5) high-pressure pump, (6) low-pressure pump, (7) oil tank, (8) suction pump, (9) suction line, (10) oil filter, (11) servo valve, (12) flow meter (upper), (13) flow meter (lower), (14) DSpace, and (15) load.

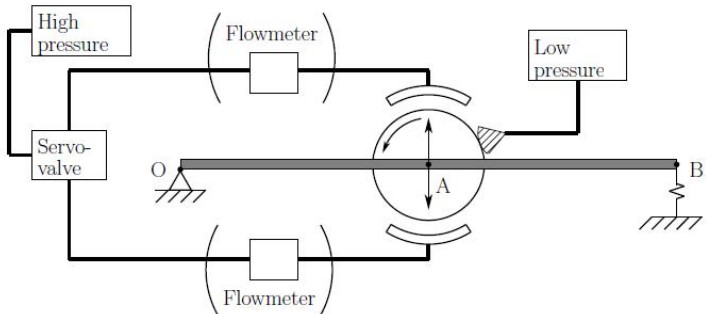

**Figure 16.** Schematic representation of the rotor-lever system.

### 5.2. Input–Output Analysis

By analyzing the input and output of each constituent mechanical component of the test apparatus, an input–output flow transformation diagram can be obtained, as shown in Figure 17, which is drawn using the concept of a SysML *internal block diagram*, which is a general modeling language for systems engineering [50]. Each block represents a mechanical component of the test apparatus. Different types of object processed by mechanical components are distinguished by color (blue, mass; red, energy; purple, force; green, signal). Objects can not only flow within a component but also be transmitted to the other components.

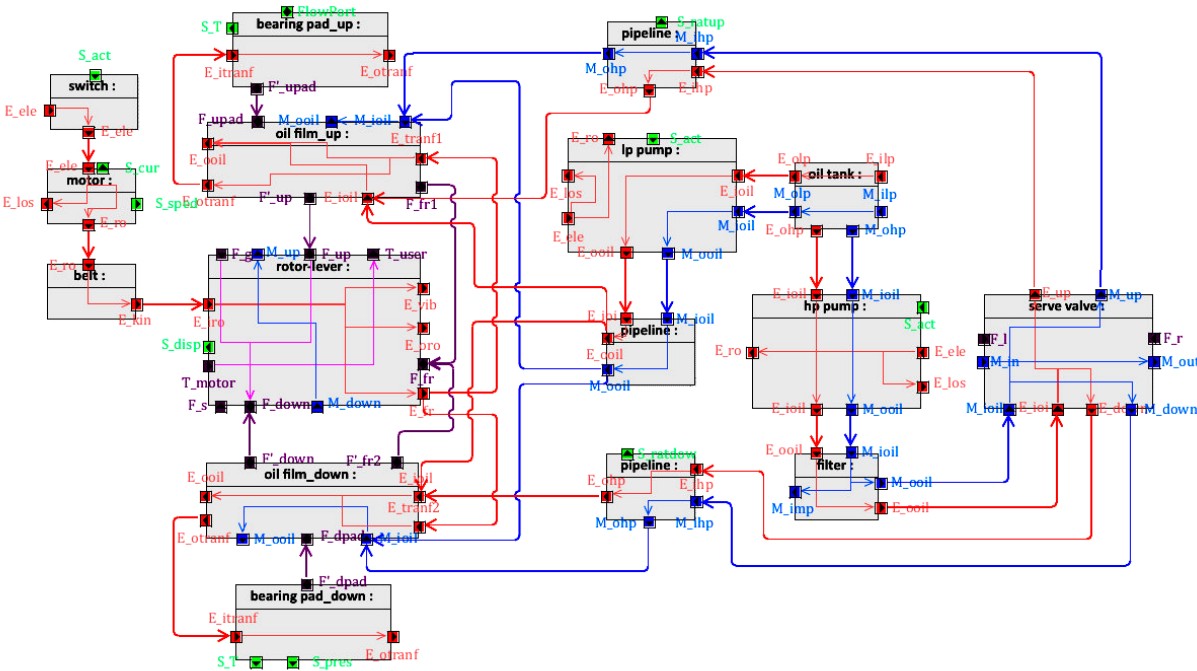

**Figure 17.** Input-output flow transformation diagram of the test apparatus.

### 5.3. MFM Model

According to the relationship between specific inputs and outputs, each mechanical component can be decomposed into several mechanical functions. Figure 18 shows the complete function model of the test apparatus, which is built on a functional modeling and reasoning workbench developed based on MFM. Mechanical functions that belong to the same component interact with one another and are clustered within a circle. Function clusters are also connected using proper relations. In this case study, information-flow functions were not specifically modeled. Status signals, i.e., condition health monitoring sensors mounted on the test apparatus, as listed in Table 2, correspond to specific MFM function primitives in the model, as marked in Figure 18.

**Table 2.** Condition monitoring data corresponding to MFM function primitives.

| Measurement | MFM |
|---|---|
| Displacement of the lever ($x_1$) | *tra*_dsplac |
| Pressure of the upper pad ($P_U$) | *tra*_presupr |
| Force of the upper pad ($F_U$) | |
| Temperature of the upper pad ($T_U$) | *sto*_tempupr |
| Pressure of the lower pad ($P_L$) | *tra*_preslor |
| Force of the lower pad ($F_L$) | |
| Temperature of the lower pad ($T_L$) | *sto*_templor |
| Flow rate at the upper pad ($q_U$) | *tra*_flupr |
| Flow rate at the lower pad ($q_L$) | *tra*_fllor |
| Rotational speed ($\Omega$) | *tra*_sped |
| Motor current | *sou*_motrelec |

The overall function of the test apparatus is to convert and transmit rotational energy to the potential user, which can be split into three subgoals of design: (1) to generate the rotational energy, which is converted from electricity; (2) to reduce friction between moving parts, which requires lubrication oil to provide clearance; and (3) to constrain the relative motion of the rotor, which requires the provision of further structural support, and to control the rotor displacement in order to ensure stability. To prevent structural failure of the bearing pad, side effects of system operation such as the accumulation of heat energy

should be reduced, which is intentionally achieved via lubrication injection. The stability is maintained by adjusting the pressure of lubrication film on the upper and lower sides through the servo valve.

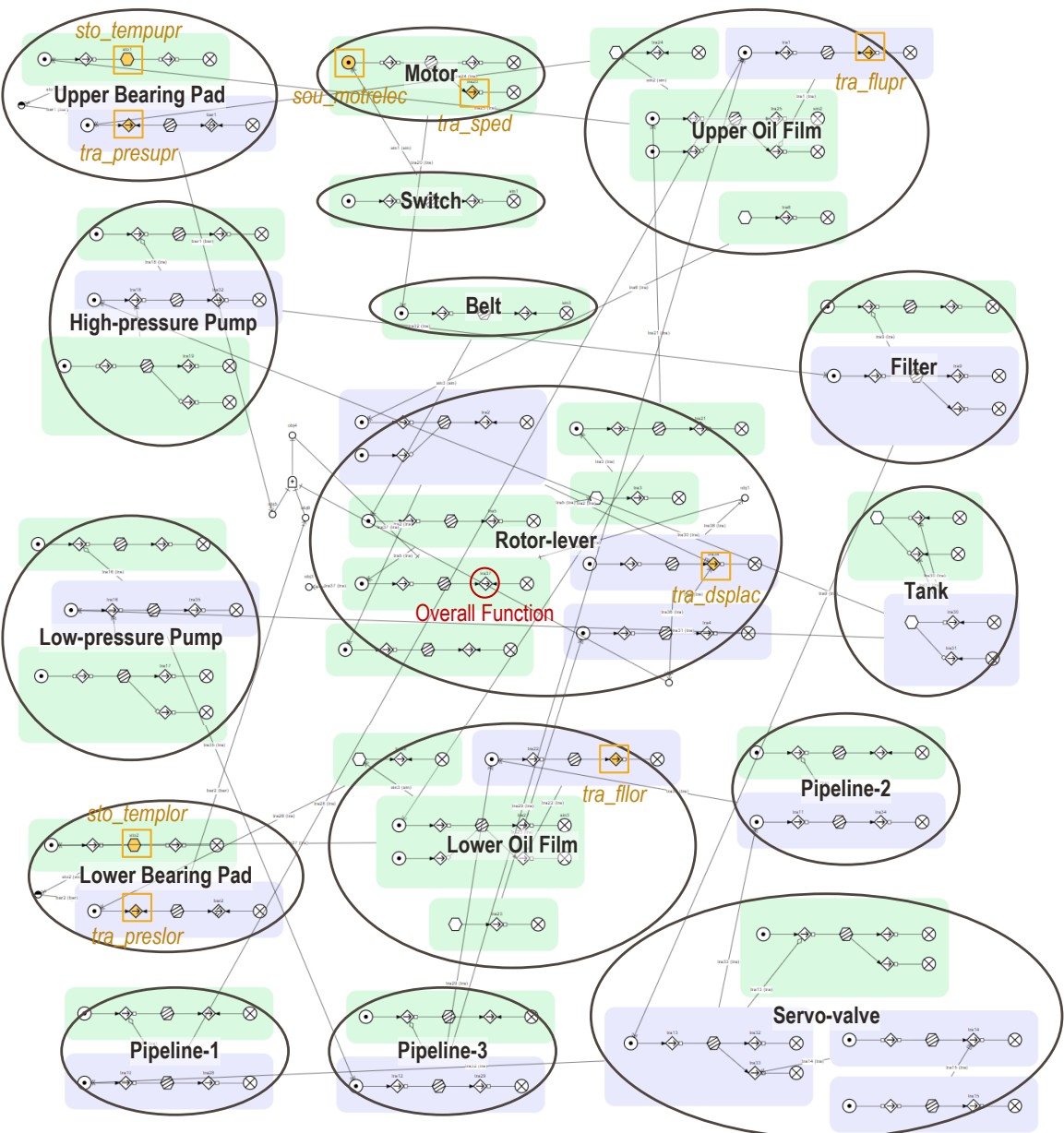

**Figure 18.** Function model of the test apparatus. Green blocks are energy-related mechanical functions, and purple blocks are mass-related mechanical functions.

### 5.4. Reasoning for Maintenance Decision Support

In order to demonstrate how function models are used for maintenance decision support along with the proposed framework, i.e., the workflow shown in Figure 11, the function model described above for the test apparatus is combined with the functional modeling of a process plant reported in a previous study. First, we describe the limitations of maintenance using the existing modeling paradigm.

#### 5.4.1. Problem Statement

Consider a deoxygenation system of a seawater injection system for an offshore platform used for oil production. The main purpose of water injection is to support the

reservoir pressure to maintain production. Deoxygenation is an important process ensuring the quality of injected sea water [44]. As shown in Figure 19, sea water is first pumped from sea by a lift pump to the fine filter, then to the deoxygenation system. Deaerated water flows to the suction of the water injection pumps after deoxygenation. Deoxygenation is primarily achieved by a physical process, i.e., stripping of the dissolved oxygen using stripping gas, and a chemical process, i.e., reaction of the dissolved oxygen with an oxygen scavenger.

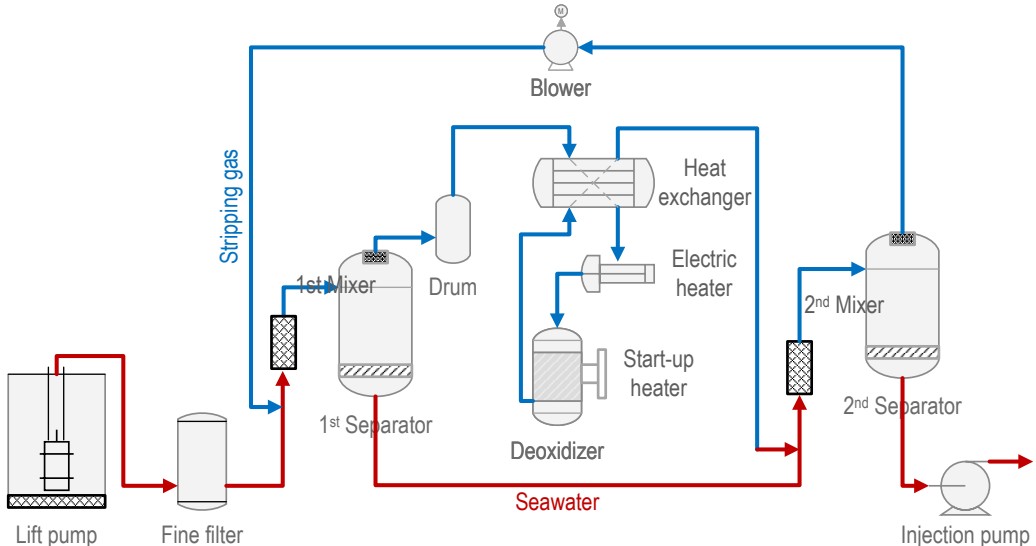

**Figure 19.** Simplified schematic of the deoxygenation system for a seawater injection system.

In a field test on an MFM-based operation support system [44], a system trip event was successfully identified, i.e., termination of water injection was caused by the functional failure of the blower, whose original function was to circulate the stripping gas flow in the loop. This event indirectly resulted in the loss of injection water for more than half a month due to the need to balance the pressure of the other injection trains. Figure 20 shows the trends of key process parameters during system stripping. For the purpose of illustration of the limitations of the existing reasoning approach, we only show how the failure of the "transport seawater" function can be traced back to the failure of the "transport gas by blower" function. The detailed process function model of the whole plant and the detailed reasoning route for the model are intentionally omitted.

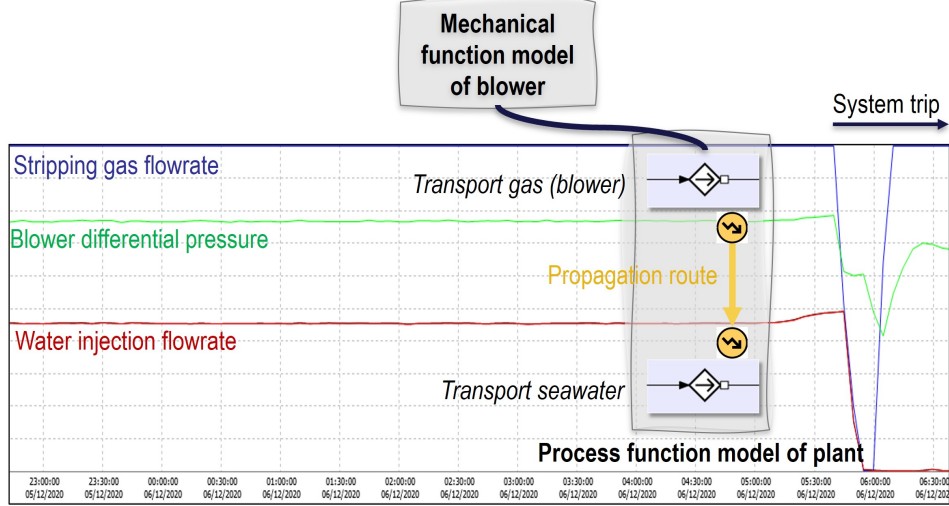

**Figure 20.** Trends of key process parameters during system stripping and the root-cause reasoning for the trip shown in a simplified function model.

Although the existing operation support system can quickly identify the root cause of a detected process anomaly and alert to the necessity of corrective maintenance, it cannot provide information about how corrective maintenance should be performed, e.g., which mechanical part has failed. More importantly, its inability to link the condition monitoring data restricts its ability to reason about mechanical failures, which hinders the possibility of preventing the occurrence of a trip. In the following section, we demonstrate how the proposed approach can extend the reasoning ability for preventive maintenance. Since in the field tests, we did not simultaneously obtain both process data and equipment condition data from the SCADA system, for the purpose of illustration, we assumed that the "transport gas" function of the blower can be connected to the function model as described before through the overall function of the test apparatus, i.e., transmitting the rotational energy.

### 5.4.2. Type 1 Maintenance Decision

First, we show how the reasoning about causes can extend to the mechanical level to enable a type 1 maintenance decision, as mentioned in Figure 11. We assume that the functional failure of the blower results from the failure of rotational energy transmission, which is used as trigger for reasoning about cause in the model shown in Figure 18. Figure 21 shows that 33 causes were identified that could possibly cause the functional failure of the blower. It should be noted that without confirmation of equipment condition data, a different approach is required to determine the 33 actual causes of mechanical failure.

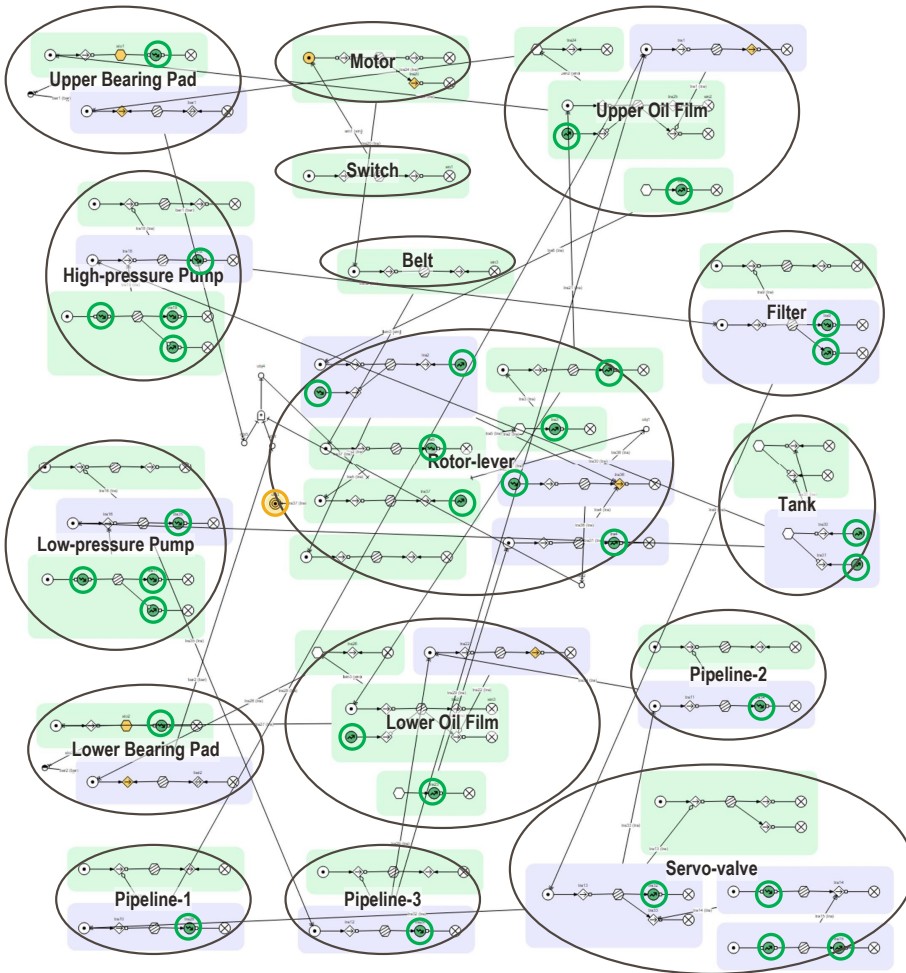

**Figure 21.** Diagnostic reasoning for functional failure of the "transmitting the rotational energy" function. The yellow circle represents the reasoning trigger. Green circles represent the identified causes.

### 5.4.3. Type 2 and 3 Maintenance Decisions

For the Type 2 and 3 maintenance decisions, we demonstrated how a detected abnormal condition of the test apparatus is used to trigger the diagnostic and prognostic reasoning in the MFM model for mechanical failure analysis. We present an experimental scenario in which the temperature of the upper bearing pad exceeds the threshold, as shown in Figure 22, to trigger reasoning about the cause and consequence. This abnormal condition corresponds to the high state of a *storage* function in the function model.

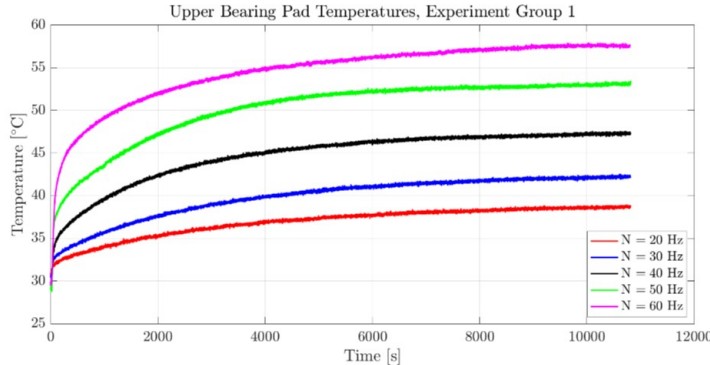

**Figure 22.** High temperature of the upper bearing pad set to trigger causal reasoning in the function model. Different colors indicate different experimental settings regarding the velocity of the rotor.

Figure 23 shows the prognostic reasoning in the function model for a hypothetical condition anomaly. The influence propagation route is highlighted. As can be seen, a high bearing temperature results in a sequence of consequences for the operation of the test apparatus. A direct consequence is that there is a risk of damaging the structural support of the bearing pad, which can result in the objective of the rotor constraint not being satisfied. As a result, the condition for transmitting rotational energy can no longer be maintained, which ultimately results in the overall function not being achieved.

During the retrospective investigation of the trip event introduced above, it was found that the temperature of wind coming out of the blower increased six days before the trip, which implies a high temperature inside the blower. This high temperature is the actual anomaly that eventually caused the trip [44]. In other words, a high temperature is critical to the operation of the water injection system (from the perspective of availability because it causes a trip), which therefore requires preventive maintenance. Therefore, it is expected that inputting the anomaly of high bearing temperature into the RCM automation system as specified in the type 2 maintenance decision will produce the same result, i.e., being critical in terms of availability. This means that run to failure is not allowed when a high upper bearing temperature is detected, and identification of potential mechanical failures that cause the detected condition anomaly is necessary.

As shown in Figure 24, 16 possible root causes were identified, each of which represents a particular influence propagation route (not shown in the figure) that can lead to a high bearing temperature. Here, root cause (RC) denotes an abnormal state of a function primitive that cannot be further caused by other failures in the model. As can be seen, some of the root causes are centered within a specific mechanical component; in such cases, each root cause can represent a specific failure mode that results in a mechanical function failure.

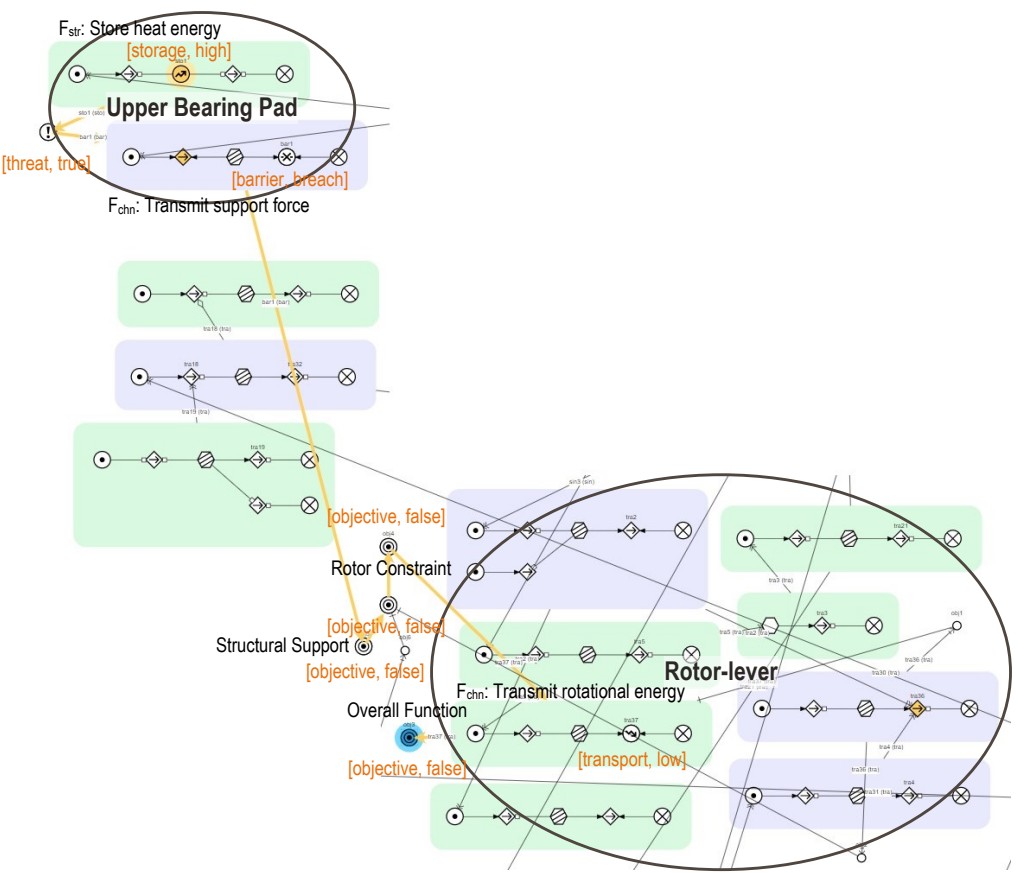

**Figure 23.** Prognostic reasoning for a high upper bearing pad temperature.

Results Evaluation

The high bearing temperature might originally result from high working motor output (RC-3), which cannot be the root-cause itself, which may also stem from a high level of electricity (RC-2) or low energy loss (RC-4). All three root causes can be seen as the failure of the "convert electricity to rotation" mechanical function. The mechanism of the condition anomaly is that this failure first transmits the high rotational energy to the belt (RC-7), which increases the friction between interfaces (RC-9), then transmits the generated heat to the bearing (RC-8) through the oil film between the rotor and the bearing. As a maintenance response to this situation, the motor speed should be adjusted.

A high bearing temperature may also result from failures of the motor of the low-pressure oil injection pump (RC-11, RC-12, and RC-13), resulting in low working output of the pump (this can also happen without failures of the motor, as represented by RC-10). Afterwards, the cooling of the bearing becomes insufficient, which can also result from pipeline problems (RC-14, RC-15, and RC-16) or a tank problem (RC-6). Therefore, relevant maintenance activity should be performed to ensure the cooling function, either by fixing the pump or inspecting the pipelines.

There are two root-causes, i.e., RC-1 and RC-5, that may not stem from failures of mechanical components but result from external reasons, e.g., a high room temperature.

Therefore, the diagnostic reasoning in mechanical function model provides three perspectives on where and how the mechanical equipment should be maintained to ensure it is able to continue operating as expected on the test apparatus, i.e., effectively transmitting the rotational energy.

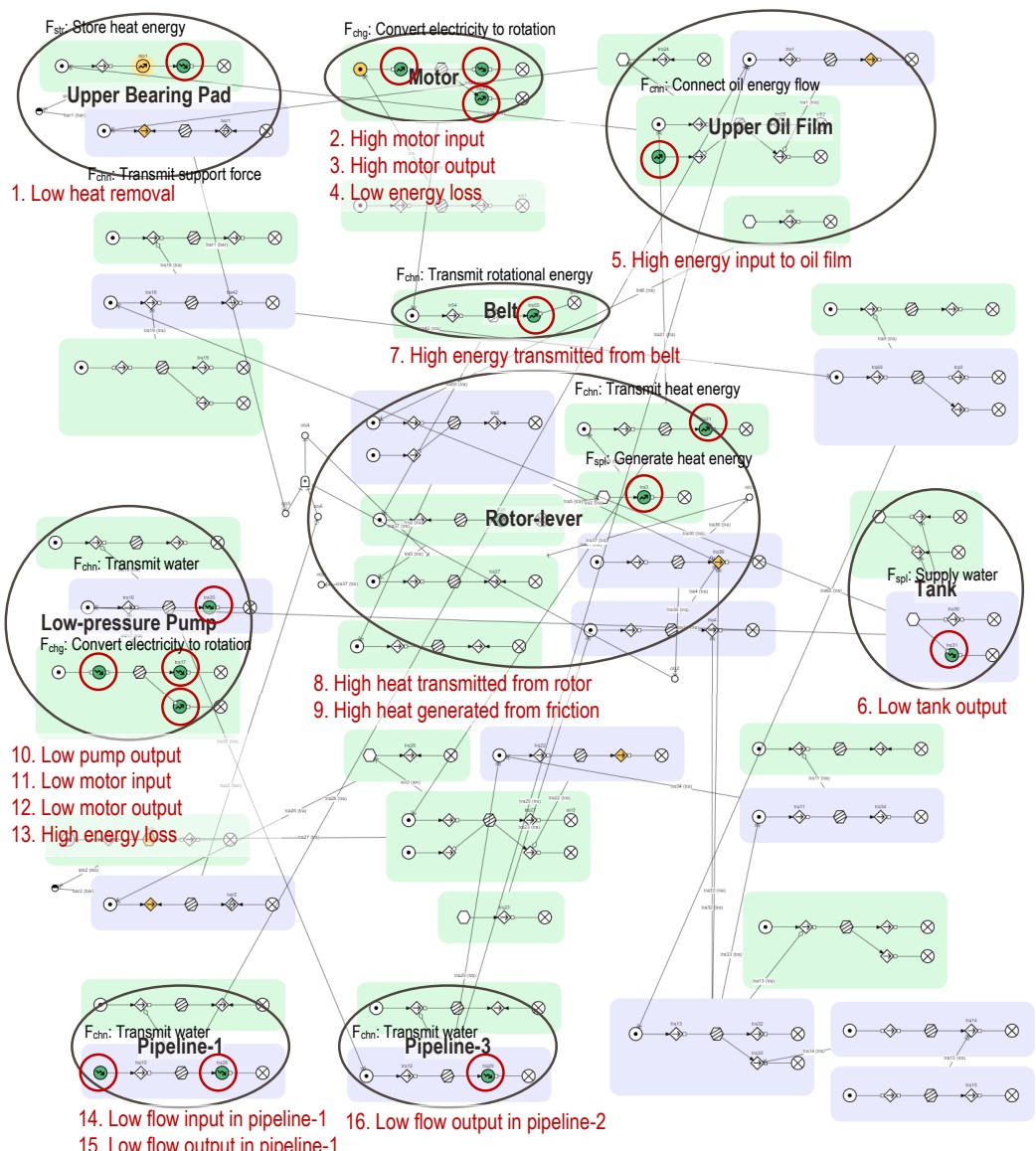

**Figure 24.** Diagnostic reasoning for a high upper bearing bad temperature.

## 6. Discussion

Representation of functions in a knowledge-based maintenance decision support system is in high demand. In comparison with CBM methods based on machine learning, functional knowledge has the following advantages: (a) it does not rely on training data and only requires basic principles of physics and existing information about how equipment is designed and operated; (b) it is consistent with how human beings understand the maintenance issue, so analysis of the results of the function model are more intuitive; and (c) functional knowledge is more general and can span levels of abstraction, which implies that it can not only be used to solve a particular equipment problem but also to align problem solving within its context, as shown in the proposed CBM framework. Functional knowledge is used for both diagnosis of failure and optimization of the maintenance strategy. Benefiting from those merits, the purpose of developing an optimized CBM such as that presented in this paper is to ensure that maintenance decisions are both cost-effective and accurate. Cost-effectiveness is realized by evaluating the functional impacts of a detected condition anomaly at the process level, which is determined by adopting an RCM automation system. Although the proposed mechanical functional modeling approach extends the ability of MFM to engage in diagnostic and prognostic reasoning with respect to mechanical failures, the accuracy of the maintenance decisions achieved

using this approach is open to dispute. Accordingly, several shortcomings of the current method need to be pointed out.

- A new functional modeling method suitable for the features of mechanical equipment was been proposed. However, we implicitly assume that the causal reasoning rules used for diagnostics and prognostics do not change after extending the modeling range, which brings up the question of how the causality between represented mechanical functions can be validated. Model validation is a common problem in MFM. For example, Nielsen et al. [51] proposed a causality validation method that also assumes that rules are invariant, despite making a comparison between causal reasoning in MFM and stochastic causality analysis in order to modify the causal relations used in the function model. Although it remains to be investigated whether new rules exist in the representation of mechanical functions, using new set of rules also increases the difficulty of developing such a rule-based reasoning system because under the unified model basis, it is difficult for a computer to determine which set of rules should be used. The solution may lie in using existing modeling syntax and semantics to adapt the newfound rules.

- There is still a gap between what the proposed method can offer and what is required by CBM. On the one hand, from the perspective of diagnostics, the potential failure identified in MFM is only implied in the corresponding component and mechanical function. How the failure is developed is not explicitly described. For example, assuming that MFM identifies "low transmit torque" as a potential failure, it is unclear how this failure is generated and how it should be eliminated by maintenance. In order to further identify the failure mechanism, failure modes, mechanisms, and effects analysis (FMMEA) [52] may be used as an additional knowledge source, which can be integrated into the mechanical functional modeling to represent possible degradation processes. In addition, ranking between different failure candidates is also required. On the other hand, from the perspective of prognostics, the method is limited to predicting the qualitative consequences rather than RUL, which is directly related to the maintenance decision. This limitation is due to the feature of instantaneous reasoning of MFM, which means that time is not considered in the influence propagation. To extend the capability of time prediction, the the temporal aspect of both functions and relations in MFM should be taken into account, allowing the consequence reasoning based on MFM not only to predict consequences of a potential mechanical failure but also to predict how long it takes to reach functional failure.

## 7. Conclusions

There is a need for a knowledge-based maintenance support method when there is a lack of training data for machine learning models. It is essential to explore mechanical functional knowledge in order to make the maintenance decision information more intuitive for maintenance personnel. In this paper, we propose a general mechanical functional modeling approach that bridges the gap between existing process functional modeling and mechanical functional modeling, essentially extending the capability of MFM from past practices in operation support to maintenance support. Moreover, in combination with the previously developed maintenance strategy optimization method, a novel framework of an optimized CBM is proposed that can offer maintenance support abilities that existing CBM methods cannot achieve. By applying qualitative cause and consequence reasoning to consistent modeling language, a function model coupling the mechanical and process levels can be used for maintenance decision support, such as (a) diagnosis of equipment functional failure for timely corrective maintenance, (b) determination of whether a detected condition anomaly of equipment needs to be further addressed by preventive maintenance in accordance with RCM, and (c) diagnosis of potential mechanical failures based on condition monitoring data.

We demonstrated the performance of the proposed framework on a mechanical test apparatus, as well as its hypothetical coupling with a process plant. Validation of the

reciprocal relation between the proposed method and existing methods regarding maintenance decision support is possible when obtaining both process data and equipment condition data from the SCADA system. As discussed, the proposed method can ensure the cost-effectiveness of CBM but may compromise accuracy. Future works should ensure accurate maintenance decisions, including validation of causality between the modeled mechanical functions, determination of real potential failures among multiple candidates, and identification of the failure degradation process. How the proposed approach can be enabled for time reasoning and associated with quantitative methods for RUL prediction also needs to be addressed.

**Author Contributions:** Conceptualization, M.S.; methodology, M.L.; software, M.S. and X.Z.; validation, X.Z. and J.W.; formal analysis, M.S.; investigation, M.S., X.Z. and J.W.; resources, I.F.S.; data curation, I.F.S.; writing—original draft preparation, M.S.; writing—review and editing, M.S., X.Z., J.W. and M.L.; visualization, I.F.S.; supervision, M.L.; project administration, X.Z.; funding acquisition, X.Z. All authors have read and agreed to the published version of the manuscript.

**Funding:** This research was funded by the Danish Offshore Technology Centre, Denmark.

**Data Availability Statement:** Data are contained within the article.

**Conflicts of Interest:** The authors declare no conflict of interest. The funders had no role in the design of the study; in the collection, analyses, or interpretation of data; in the writing of the manuscript; or in the decision to publish the results.

## Abbreviations

The following abbreviations are used in this manuscript:

| | |
|---|---|
| CBM | Condition-based maintenance |
| CPD | Causal process description |
| MFM | Multilevel flow modeling |
| MeFM | Mechanical function model |
| RCM | Reliability-centered maintenance |
| FBD | functional block diagram |
| FMMEA | Failure modes, mechanisms, and effects analysis |
| PHM | Prognostics and health management |
| PrFM | Process function model |
| RC | Root cause |
| SCADA | Supervisory control and data acquisition |

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
