# Peer review of "Explicit Representation of Mechanical Functions for Maintenance Decision Support"

_electronics, doi:10.3390/electronics12204267_

Round 1

Reviewer 1 Report

This manuscript proposes a (1) mechanical functional modeling approach based on multilevel flow modeling (MFM), which provides functional modeling and reasoning methodology, and (2) a framework of CBM combining MFM and RCM. MFM has been used for functional modeling, but has not been connected to maintenance. This gives the novelty in the proposed methodology. 

Overall, this reviewer saw and agreed on the relevance and importance of the described gaps and the contribution of the proposed approach. However, there are two main feedback which need to be addressed. (1) Explanations of the modeling approach, figures, and application procedures could be more precise. The specific questions of this reviewer are given below. (2) Some contributions the authors argue need to be more clearly stated or supported through the case study on the rotor-lever example.

Specific questions regarding the unclear explanations:

-     First of all, the explanation of the modified MFM is a bit vague for a reader who is not familiar with traditional MFM. It feels like many steps have been omitted.

-     Figure 3 is not self-explained for this reviewer. Some elaboration is necessary, such as the goal/function/means/end, why there are four means-end pairs (are they an example?), what is part-whole, where are input/output in Figure 3, etc. These might be unclear only because this reviewer is unfamiliar with MFM. However, the manuscript can give a brief introduction as MFM is intensively used in this work.

-     The notation of functions is unclear for this reviewer. \sum_{i=1}^{m} seems to be the union set of elements (input/output), not the summation of values. Also, what is < , >? Does this indicate a pair of two sets? I wonder whether these are conventionally accepted notation.

-     Figure 6 does not stand alone. Readers need some guidance and explanation on how to read/understand the figure.

-     Line 304, it is unclear what is “vary”, and how this is connected to the patterns in Figure 6.

-     Line 306, where can I find “mechanical function channel” in Figure 6?

-     Line 546-547 and Figure 10, there is no “Type-2 maintenance decision” in Figure 10.

Specific questions regarding less convincing contributions:

-     The proposed CBM framework is posed as “optimized” CBM. However, there is no optimization in the described framework. What is actually optimized in this framework? What is the criteria/objective/goal of the “optimality”? How do you show/guarantee that the proposed framework (or the actions suggested by the proposed framework) is “optimal”?

-     Line 385, what do you mean by “accuracy” in maintenance? Performing the right maintenance action for the current system condition? Performing at the right moment just before the system failure? Preventing any failure? What does it mean?

-     Line 546-547, connecting RCM to CBM has been described as one of the main contributions of the proposed CBM framework. Then, this should be illustrated through the case study.

Other questions and comments:

-     In Figure 10 and the framework, what is process-functional-anomaly? Why do we run to failure for non-process-functional-anomaly? Do we always run to failure for them? 

-     Figure 13, can we see a bigger picture of the diagram, so that the diagram and the photo can be compared?

-     Line 523, what do you mean by “temperature accumulation”? Is the temperature something accumulated? Do you mean high temperature?

Some minor typos:

Line 298, connected by balance, with (which) has the function of change.

Figure 7, there are two P_up. One must be P_down.

Overall, the English was not a problem while reading this manuscript. The main improvement should be in the explanation, not the English itself.

Reviewer 2 Report

Minor issues

Old references are used to contextualize current aspects (for example: 1, 18).

The statement that machine learning algorithms are intrinsically black box should not be made. Many machine learning methods are fully and easily interpretable.

The work talks about cost-effectiveness without analyzing the point of view of asset management, an approach currently widely used in maintenance management.

Regarding the case study:

The approach using the criticized methods should also be carried out and its results compared to the results of the proposed method. The comparison could, or should, encompass:

- costs (effort, man-hour, computational cost, etc.);

- technical assertiveness.

The case study did not conduct diagnoses and prognoses as proposed in the method, it was superficial.

Regarding the Discussion

Artificial intelligence is seen as a competitor to the proposed method and more laborious as it requires training data.

Notably, artificial intelligence could be complementary to the proposed method: the operation and maintenance of the equipment begins with the proposed method. Over time, maintenance management can be improved through artificial intelligence based on real operation and maintenance data.

The claim that machine learning is an uninterpretable black box should not be made. Only deep neural networks and the like are not interpretable. Regressions, simple artificial neural networks, for example, are fully interpretable.

The statement that artificial intelligence would work for only particular cases should not be made. It learns based on the data presented to it. Therefore, if there is data, there will probably be learning in the dimension that the data represents.

Artificial intelligence being capable of modeling that cannot be interpreted by humans may even be unethical, but when used where humans cannot model, it is not a competitor, it is complementary.

Reviewer 3 Report

Thank you for giving me the opportunity to read this paper. The authors made extensive research and put considerable effort into it. The study is adequate and diverse, interesting results are presented and discussed. However, some improvements could be made before this manuscript is published. Comments are listed below.  

1.       "Literature Reviews" should be divided into one independent section.

2.       I found the paper to be well explained and well written, but it is difficult to follow the overall structure of the article. I suggest that you 1) add a figure showing the entire framework of this research to highlight the model development procedure, and 2) explain it in detail.

3.       Contributions in the "7. Conclusion" section must be more precisely indicated. The theoretical contribution and practical implications of this manuscript should be clearly stated.

4.       It is recommended to mention this study's limitations and future works in the Conclusion section.

5.       minor comment - In Figure 17, pipeline-1 and pipeline-3 overlap with the text in the figure. Please separate the text within the figure so that they do not overlap.

Round 2

Reviewer 2 Report

I thank and congratulate the authors for the quick response and the clarifications in the text.

For a scientific journal article, I recommend that the evaluation of the proposed method be more comprehensive, enabling a comparison with the results of methods currently in use.

It would be possible, for example, to evaluate a system that has been in operation for some time, applying the proposed method "in the past", checking how its performance would be and comparing the results with the actual performance that the system had.
